# Comparative Analysis of Spectroscopic Studies of Tungsten and Carbon Deposits on Plasma-Facing Components in Thermonuclear Fusion Reactors

Vladimir G. Stankevich *, Nickolay Y. Svechnikov and Boris N. Kolbasov

National Research Centre "Kurchatov Institute", 123182 Moscow, Russia
* Correspondence: vl-stan@mail.ru

**Abstract:** Studies on the erosion products of tungsten plasma-facing components (films, surfaces, and dust) for thermonuclear fusion reactors by spectroscopic methods are considered and compared with those of carbon deposits. The latter includes: carbon–deuterium $CD_x$ ($x \sim 0.5$) smooth films deposited at the vacuum chamber during the erosion of the graphite limiters in the T-10 tokamak and mixed $CH_x$-Me films (Me = W, Fe, etc.) formed by irradiating a tungsten target with an intense H-plasma flux in a QSPA-T plasma accelerator. It is shown that the formerly developed technique for studying $CD_x$ films with 15 methods, including spectroscopic methods, such as XPS, TDS, EPR, Raman, and FT-IR, is universal and can be supplemented by a number of new methods for tungsten materials, including in situ analysis of the MAPP type using XPS, SEM, TEM, and probe methods, and nuclear reaction method. In addition, the analysis of the fractality of the $CD_x$ films using SAXS + WAXS is compared with the analysis of the fractal structures formed on tungsten and carbon surfaces under the action of high-intensity plasma fluxes. A comparative analysis of spectroscopic studies on carbon and tungsten deposits makes it possible to identify the problems of the safe operation of thermonuclear fusion reactors.

**Keywords:** spectroscopic methods; plasma–surface interactions; carbon–deuterium smooth films; tungsten erosion products; fractal structure



## 1. Introduction

Our previous work using a number of techniques to study carbon–deuterium $CD_x$ films formed in the plasma discharges of a T-10 tokamak [1,2] was due to the general problem of the interaction of plasma particles with the first wall in a thermonuclear fusion reactor, the sputtering of wall materials (in this case, carbon) and the formation of films and dust that absorb the operating gas in the form of H, D, and T isotopes used in fusion devices.

The relevance of the research is also associated with the construction of the novel JT-60SA fusion reactor (Super Advanced, Naka, Japan) with superconducting magnetic coils made of NbTi and $Nb_3Sn$ alloys, and a liquid helium cooling system [3]. It is a modification of the JT-60U tokamak, and it is being built jointly with EU countries; the first wall and divertor are made of carbon fiber composites, which will be used during the first 10 years after launch [4]. Its creation is carried out in support of the ITER and DEMO projects; therefore the JT-60SA reactor is called "a satellite tokamak".

The original design agreement between Japan and the EU stated that carbon plasma components were the most reliable for producing long-pulsed H-mode plasma (high-confinement operating mode, with the energy confinement time significantly enhanced) at 5.5 MA with high β (ratio of the plasma pressure to the magnetic field pressure) at a full injection power of 41 MW for a 100 s pulse duration, with high electron and ion plasma temperatures ($T_i$ and $T_e$ = 6.3 keV) and a high plasma density ($\sim 1 \times 10^{20}$ m$^{-3}$). At a later

stage, the divertor and the first wall will be completely replaced with tungsten-coated carbon material.

As noted, the reason for the carbon choice is the rich experience in working with carbon materials, low contribution to the radiation losses of the central plasma (i.e., the plasma temperature does not decrease), high thermal stability, wide operating temperature window, high resistance to heat flux, and high thermal conductivity. According to [5], carbon remains the main component of the material in most of the existing experimental devices, but "both low and high Z choices for plasma facing surfaces in present tokamak fusion devices have significant performance issues".

Carbon may be present in the metal first wall tokamaks as impurities from previous carbon wall campaigns (as will be shown here in the JET and the mixed $CH_x$—Me films formed by the QSPA-T high-current coaxial plasma accelerator) or remnant deposits after studies of carbon-containing materials, as well as from carbon deposits in the form of atmospheric gases. Carbon will be used in the first stages of the JT-60SA's operation until a gradual transition to all-metal walls (for a period of at least three years) and the subsequent periodic replacement of worked–off plasma components using a remote maintenance system [6]. Prior to the launch of ITER, it will be the most powerful tokamak, only half the size of ITER.

Among the stated goals of the JT-60SA is to complement the research being carried out to justify the construction of the ITER reactor, as well as to create a technological basis for the development of a demonstration fusion reactor (DEMO).

Since tungsten-coated carbon fiber composite materials are attractive for experiments with a "metal wall", it is necessary to prepare a technology for their manufacture. For example, a method is being developed for sintering thick tungsten plates and blocks of carbon fiber composite [7]. Reports present the research on the preparation of W/CFC compounds, their main micro-structural and mechanical characteristics, and the results of preliminary tests for high-temperature exposure at 1073 and 1173 K for 1000 h in vacuum conditions. It is also planned to test an ultrafine-grained W-TiC alloy.

At this stage, the carbon first wall is used in modern divertor-type tokamaks: DIII-D (the largest tokamak in the USA, San Diego, with a high β = 12.5%), NSTX-U (National Spherical Torus Experiment Upgrade, USA, NJ, Princeton, with an in situ material analysis system MAPP–materials analysis particle probe), and KTM (materials science tokamak launched in Kazakhstan). For example, the NSTX-U at the Princeton Plasma Physics Laboratory is testing various candidate plasma contact materials, including low Z-coated graphite and high-Z materials (W and Mo). A program for the lithiation and boronization of plasma materials is also being conducted [5].

These tokamaks are among the 35 currently operating [8], where experiments are carried out to study and control high-temperature plasma for the purposes of ITER, including such tokamaks where the configuration of the magnetic field on a reduced scale repeats the configuration of the arrangement of the coils for controlling the poloidal magnetic fields in ITER. These are ASDEX Upgrade (Germany, Garching), Tore Supra/WEST (France, Cadarache), EAST (Experimental Advanced Superconducting Tokamak, Hefei, China), KSTAR (Korean Superconducting Tokamak Advanced Research), and in Russia these include a hybrid tokamak, T-15MD (Moscow, Russia, National Research Center "Kurchatov Institute"), and the Globus-M spherical tokamak (St. Petersburg, Russia, A.F. Ioffe Physical Technical Institute).

The usefulness of the methods for studying $CD_x$ films in other thermonuclear experiments within the framework of the ITER project was discussed in [9,10] in relation to the study of erosion products of various types of first wall coatings. For example, they may include C–Me mixed films, such as the $CH_x$–Me films from the QSPA-T plasma accelerator [11] considered in this work, as well as metal coatings such as Be, W, and Fe, and their alloys.

In this work, the primary attention is paid to examples of modern plasma–surface studies on tungsten-containing and carbonaceous erosion products of the tokamak first

wall, along with a comparison with similar studies on carbon erosion products in the form of $CD_x$ films and mixed $CH_x$–Me films. The emphasis is made on demonstrating the possibility and efficiency of studying the erosion products of the first wall materials by spectroscopic methods in order to obtain new physical results.

In particular, spectroscopic methods prevailed in the study of $CD_x$ and $CH_x$–Me hydrocarbon films, and they showed their perspective potential for the study of tungsten-containing materials, along with the new methods considered in the works by other authors.

Carbon and tungsten are unique initial wall materials with different types of lattice symmetry, and their erosion products show different lattice symmetries, arising under the influence of plasma fluxes of different types. Thus, in graphite wall material, the layers of the crystal lattice can be arranged differently relative to each other, forming a number of polytypes, with symmetry from hexagonal to trigonal syngony. Tungsten with a stable modification ($\alpha$-W) forms crystals of the body-centered cube (bcc). Primitively, the most common erosion products of tungsten are known to be $WO_3$, $WO_2$, and WC. However, one of the main products of tungsten erosion, tungsten trioxide $WO_3$ with a high valence $W^{6+}$, leads to the tilting of the neighboring $WO_6$ octahedra and antiferroelectric displacement of atoms due to the strong electrostatic interaction between $W^{6+}$ ions. Similar crystallographic distortions are responsible for five phase transitions, which occur depending on the temperature: stable monoclinic phase—down to $-27\ °C$; triclinic in the temperature range from $-27$ to $20\ °C$; monoclinic from $20$ to $339\ °C$; rhombic from $339$ to $740\ °C$; tetragonal from $740$ up to $1470\ °C$. Tungsten dioxide $WO_2$ possess a rutile-like (monoclinic) structure with distorted octahedral $WO_6$ centers with alternate short W–W bonds and a $d^2$ electronic configuration, which imparts the material a high electrical conductivity. Tungsten carbide WC has two forms: a hexagonal form of $\alpha$-WC and a cubic high-temperature form, $\beta$-WC, which has a rock salt structure [12].

Moreover, it was established that the considered so-called film-type "disordered" erosion products of carbon and tungsten from plasma high flux fusion devices do not exhibit a trivial disordered structure but show fractal symmetry at different levels of the spatial scale. At the same time, the hierarchy of the structures (surface or volume) is described by a power law with different fractal dimensions depending on different plasma exposure conditions, where for linear surface structures it extends from subnanometer sizes to ~10 μm.

The spectroscopic methods often used in the study of erosion products are X-ray photoelectron spectroscopy, infrared spectroscopy, and Raman spectroscopy. The high efficiency of the Raman spectroscopy of tungsten oxides, which constitute the main fraction of tungsten deposits, is provided by a much richer Raman spectrum than that for amorphous $CD_x$ films, e.g., the most abundant $WO_3$ deposit has a quasi-orthorhombic structure with the P21/n symmetry and, therefore, a strong electron–phonon coupling inherent to perovskites.

At the same time, the technique for studying $CD_x$ films, as will be shown, has a universal character, and a number of new methods for studying the erosion of tungsten-containing materials, including the use of unique in situ analysis methods, complement and enhance it. The importance of obtaining new physical results with a tungsten first wall was also noted, which makes it possible to reveal new material properties and related problems that complicate the safe operation of fusion devices.

Finally, in this work, the authors omit the important problem of using liquid-metal protective materials (especially Sn and Li [13]) of the divertor as an alternative to the conventional solid metallic materials considered here. Only one remark should be done noted in favor of the universality of the XPS method when looking for traces of T-10 lithiation. Traces of lithium were first detected in the XPS spectra of $CD_x$/Si(100) thin films in preliminary experiments to develop the technique of lithium protection of the T-10 tokamak graphite limiter [9]. During discharges, not only the formation of $CD_x$ films took place, but also the sputtering of the lithium deposit on the chamber walls occured, which was registered as Li impurity in the $CD_x$ films. This was followed by intensive work on

lithiation with the already tested spectroscopic methods of investigation [14,15]. In general, liquid lithium protection of the first wall is a separate novel promising area of research, which also ensures the safe operation of fusion reactors.

## 2. Materials and Methods for Carbon Erosion Products from a T-10 Tokamak and QSPA-T Plasma Accelerator

We studied the $CD_x$ films formed in a diverterless tokamak T-10 (National Research Center "Kurchatov Institute", Moscow, Russia), which is a magnetic confinement device for axisymmetric toroidal thermonuclear deuterium plasma with a 0.5 MA discharge current and a 1 s pulse duration, with the dimensions of a large torus radius of 1.5 m, a small radius of 0.39 m, a longitudinal magnetic field of 2.8 T, a movable limiter and a fixed annular diaphragm (annular limiter) made of fine-grained MPG-8 pyrolytic graphite, and designed to limit the region of the central plasma and protect the chamber walls from thermal load [1].

Samples of the main four types of films were obtained for study as erosion products under the two various experimental conditions, namely, at the vacuum chamber's first walls of the T-10 tokamak and at material surfaces irradiated with plasma of the QSPA-T coaxial high-current plasma accelerator for particle and radiation heat fluxes typical for ITER transient events [16].

Namely, the following samples were analyzed:

1.  Free-standing (without a substrate), ~20–30 μm thick, carbon–deuterium $CD_x$ ($x = D/C \approx 0.3$–0.8) films with a high atomic content of hydrogen isotopes and an area of ~0.5 $cm^2$ each. They were formed in approximately 1600 electric discharges in deuterium gas, and in cleaning discharges with low-temperature plasma with a total duration of approximately 1000 h. The films exfoliated from the inner walls of the vacuum chamber, with an increase in thickness and the appearance of internal stresses. In general, these films are deposited and re-deposited on the walls of the vacuum chamber "in the shade" from the direct action of the central plasma, at an average wall temperature of 300–400 K, and become the main parasitic accumulator of hydrogen and hydrocarbon isotopes. Samples may be scraped from the chamber walls, turning into free-standing films, or flakes. Flakes have different colors from dark-brown to goldish and yellow, and the color difference is clearly related to the variable hydrogen concentration. For the most studied films of a reddish-golden color, the atomic concentration was $D/C \approx 0.5$–0.8 ($H/C \approx 0.1$–0.2, which is determined mainly by the storage conditions in the atmosphere), and for those that were dark-brown and blue–brown—, $D/C \approx 0.3$–0.4. These flakes were studied using all the below listed experimental methods 1–15.

2.  Thin $CD_x$ films (100–500 nm) were deposited on polished metal mirrors used to transmit optical plasma radiation to detectors and located at a certain distance from the central plasma under controlled deposition distances. They were exposed to radiation that came directly from the central plasma, i.e., these films were optically thick with a typical ratio of $D/C \sim 0.2$–0.4 in the case of room temperature deposition. The research methods used were Fourier-transform infrared reflection spectroscopy and X-ray photoelectron spectroscopy.

3.  Thin $CD_x$ films (100–700 nm) on Si(100) substrates inserted into vacuum at special plates in the port-plugs at the level of the chamber wall surface and irradiated in campaigns in 2010 and 2011 under special deposition conditions. These included exposure to either stable discharges (~200 discharges), discharges with plasma disruption, or cleaning discharges of deuterium low-temperature plasma. Research methods: X-ray photoelectron Spectroscopy and X-ray auger electron spectroscopy (survey spectrum, Auger CKVV electrons, C1s, O1s bands, valence band, impurities, Fe2$p$, etc.). This type of samples also included the model analogues of tokamak-produced $CD_x$ thin films, which were obtained in a magnetron, namely, a-C:H(D) polymer films formed

on metal mirrors. Research methods: X-ray photoelectron spectroscopy, X-ray auger electron spectroscopy, Raman spectroscopy, and ellipsometry.

4.  Mixed hydrocarbon films $CH_x$—Me, with metal impurities (Me = W, Fe, etc.) up to 5 at. %, and 0.4–2 µm thick on a Si(100) substrate, obtained on a plasma accelerator QSPA-T by sputtering graphite and tungsten targets (pure W = 99.95%). The high-current plasma accelerator QSPA-T, adapted for materials for ITER, used $H^+$ hydrogen 0.5 keV pulsed plasma flux with a 0.5 ms pulse duration at 50 pulses, $10^{22}$–$10^{23}$ $m^{-3}$ plasma density, forming a high heat load on the W target surface of ~0.5 MJ/m² [16]. The tungsten plate was located on the plasma flux axe with a normal at an angle of $60°$ to the plasma flux. The Si(100) substrate was installed in front of the W-target on the wall of the target chamber at a distance of ≈20 cm from the W target. Note that the materials' deposition rate during the plasma pulsed processes can be much higher than that for a stationary regime. Research methods: X-ray photoelectron spectroscopy and X-ray auger electron spectroscopy (survey spectrum, C1s, O1s, valence band, Auger CKVV electrons, $WO_3$, impurities, etc.).

Recall the research methods (for example, [1,2,17]) used for the $CD_x$ films' study (see Table 1), including the excitation energy range used in these methods and the approximate depth of the radiation penetration into carbon (for a density of ~1 g/cm³).

**Table 1.** Experimental methods used for the $CD_x$ films' study.

| № | Experimental Methods | Excitation Energy, eV | Penetration Depth in Carbon, µm |
|---|---|---|---|
| 1 | Fourier-transform infrared (FT-IR) reflection spectroscopy | ~0.02–0.5 | ~1–10 |
| 2 | Raman spectroscopy | ~2–3 | ~0.1 |
| 3 | Optical spectroscopy (OS) and photoluminescence (PL), including those using synchrotron radiation (SR) | ~2–3 | ~0.1 |
| 4 | X-ray photoelectron spectroscopy (XPS) | 1486.6 | ~0.01 |
| 4a | X-ray auger electron spectroscopy (XAES) | ~250 | ~0.001 |
| 5 | X-ray absorption spectroscopy—EXAFS FeK-edge andNEXAFS C1s CK-edge spectroscopy using SR | ~7100 ~290 | ~800 ~1 |
| 6 | Knudsen thermal desorption mass spectrometry (TDS) (using Knudsen evaporative cell), up to T = 1300 K | ~ 0.1 | ~$10^6$ (1 cm) |
| 7 | Electron paramagnetic resonance (EPR) spectroscopy (9.90 GHz, X-band, λ = 3 cm) with field scanning at 100 and 6000 Gauss | | ~$10^6$ (1 cm) |
| 8 | X-ray fluorescence analysis using SR (XRFA-SR),(as an analog of the energy-dispersive X-ray analysis (EDX)) | ~20,000 | ~$10^4$ |
| 9 | X-ray diffraction (XRD) using SR | ~8000 | ~$10^3$ |
| 10 | Small-angle and wide-angle elastic X-ray scattering (SAXS + WAXS) using SR | $10^4$–$3 \times 10^4$ | 3000–10,000 |
| 11 | Neutron diffraction (elastic neutron scattering) | ~0.025 | ~100 |
| 12 | Current-voltage characteristics | | |
| 13 | Thermogravimetric analysis (TGA), up to T = 1300 K | ~0.1 | ~$10^6$ (1 cm) |
| 14 | Nuclear methods involving light nuclei to estimate the content of H(D) (Rutherford backscattering of hydrogen or helium ions and spectrometry of recoil nuclei using helium ions of ~MeV energies) | ~MeV | ~cm |
| 15 | Theoretical modeling—for evaluating the TDS results, including the model of hopping diffusion of H(D) atoms for a wide TD spectrum region with a maximum near 750 K [16], as well as estimating the role of iron catalysis in lowering the TD barrier [18,19] | | |

Additional information on the radiation penetration depth into carbon and tungsten for different methods will be shown further within the text.

Firstly, the first eight methods in Table 1 are related to spectroscopy, which is also widely used in the case of tungsten-containing materials, as will be shown below.

Secondly, attention should be paid to the presence in the list "deeper" (by the thickness of the samples and in the depth of the radiation penetration) or bulk research methods, which include the following methods: 5–11, 13 and 14 in this list.

Thus, the X-ray absorption used in methods 5 (EXAFS FeK-edge, excitation energy $h\nu$ = 30 keV), 8 (XRFA-SR, excitation energy $h\nu$ = 21 keV), 9 (XRD-SR, ~10 keV), and 10 (SAXS + WAXS, ~10–26 keV) penetrates into the carbon material to a depth of 3000–10,000 μm and into tungsten—up to 7–20 μm, as in the case of neutron diffraction 11 and nuclear methods 14.

The microwave radiation (method 7) used in EPR 9.9 GHz, or wavelength $\lambda$ = 3 cm, penetrates into dielectrics to depths comparable to that of the wavelength.

Here, for comparison, an impurity analysis by EDX spectra (Table 1, method often used in the cited papers), (i.e., when excited by an electron beam with energies up to ~25 keV of the characteristic X-ray radiation of impurities), provides an order of magnitude lower penetration depth into carbon and other materials compared to the X-ray fluorescence analysis used for the $CD_x$ films, associated with the excitation of X-ray radiation ~20 keV from an SR source.

Thermal desorption Knudsen mass spectrometry, method 6, makes it possible to register gases desorbed from the entire sample, as well as the TGA method 13, controls the phases of the mass change in the entire sample at a certain temperature during thermal heating.

Thus, these methods using X-ray excitation and nuclear reactions involving light ions penetrating deeper into the sample, as well as EPR (also applicable to $WO_3$ erosion product) can also be used when working with tungsten up to a thickness of ~10 μm, while for neutron diffraction and the TDS and TGA methods—to a depth of tens of microns or more.

To study thicker (cm) layers one may use neutron activation analysis, which is widely used in metallurgy (impurities search), geology, biology, etc. When irradiated with thermal neutrons from a reactor or from a neutron source (such as the radioactive decay of $^{252}$Cf), the identification of the desired elements occurs using nuclear $n\gamma$ reactions.

### 3. Results and Discussions

*3.1. Research on Carbon Erosion Products from T-10 and QSPA-T*

The above methods were used to conduct comprehensive studies of the main properties of $CD_x$ films ($x \sim 0.5$) with a high atomic content of hydrogen isotopes, depending on the types of plasma discharge.

This included the study of the structure, chemical composition and micro-impurities, the electronic structure of the valence band, C1$s$ and O1$s$ core levels and impurities, vibrational modes (C–D and C–H) of hydrogen isotopes, photoluminescent properties, surface electrical properties, adsorption states of hydrogen isotopes and mechanisms of thermal desorption, and spin states of carbon and impurities of $d$-metals (arising from erosion of the tokamak chamber walls) [1,9,10,17–19].

In addition, included is a comparison of the optical properties and vibrational modes of $CD_x$ and a-C:H(D) films on diagnostic mirrors, features of the electronic structure of thin films under controlled deposition conditions on a T-10 tokamak, and a comparison with the electronic structure of hydrocarbon films from a plasma accelerator QSPA-T with a high intensity of hydrogen plasma fluxes, the effect of Fe impurity on a decrease in the thermal desorption threshold, and on the $sp^3 \rightarrow sp^2$ conversion upon heating (the so-called "iron catalysis") [11,18,19].

Due to the XPS data, distinctions of the electron structure on both sides of flakes, ~20–30 μm thick (concave-shaped plasma side of the film and convex-shaped wall side, touching the vacuum chamber), were found [10]. This indicates some non-uniformity of the $CD_x$ flake electron structure throughout the film thickness on both sides: plasma side and wall side. This refers to the elemental composition, the relative concentration of elements

and their chemical states, and the C$sp^2$/$sp^3$ ratio. Differences in the electronic structure and the vibrational properties were found between both sides of the ~30 nm thick films, as well as differences in the properties of the films from the working plasma discharges with and without the disruption of the discharge electric current, and from the cleaning discharges, which were due to the different plasma processes of their formation in tokamaks. For example, on the plasma side, due to the additional reaction $C_xH_y$ + H (H is the plasma component), a disruption of the double bonds of the sp$^2$—hybridized hydrocarbons takes place, followed by the addition of H or D atoms and the transformation of the sp$^2$ state into the sp$^3$ one, with the ratio of these states being higher for the plasma side. From the side of the wall, the adsorption of metal impurities into the carbon system leads to the breaking of chemical bonds and an increase in the formation of the sp$^2$ structure (the catalytic effect of transition metals) [10].

As a result, these properties of the films are, in fact, a "passport" to the processes of their formation under the plasma conditions of the T-10 tokamak [9].

The fractal structure of smooth amorphous $CD_x$ films was established using SAXS + WAXS as effective methods to study the structure of disordered materials. The structure consists of mass fractals with a rough boundary, surface fractals with a rough surface, flat scatterers, and linear chains forming a branched and highly crosslinked three-dimensional carbon network [2]. The fractals, including sp$^2$ clusters, are of the typical size ~1–60 nm.

All of these studies, as in most similar cases on tokamaks, were carried out ex situ, i.e., after removing the samples from the tokamak vacuum chamber following exposure to a large number of discharges. Several new results are shown further in the text.

These methods, as well as those described below, are also promising in experiments on tokamaks with tungsten-containing erosion products.

However, to carry out such studies in situ directly on tokamaks, unique developments of appropriate plug-in devices and mechanisms are required, which is currently implemented in practice only for a number of techniques and only in a few tokamaks and plasma devices (see below).

### 3.2. Approximate Estimate of the Adsorbates' Accumulation: The Relationship of the Color and Electron Structure

#### 3.2.1. The Relationship of the Color and Electron Structure in $CD_x$ Flakes

An approximate estimate of the atomic abundance of hydrogen isotopes $x$ = D/C in $CD_x$ films without resorting to more complex nuclear physics methods can be made from the band gap value $E_g$ estimated from the valence band spectrum in the XPS method. The value of D/C is also related to the color of the film, which can be estimated by taking photoluminescence spectra [20]. By measuring the $E_g$ value and using the approximate empirical linear relationship between $E_g$ and the [H] concentration, obtained empirically for soft hydrocarbon a-C:H films, though in a limited range of atomic concentrations [H] = 20–50 at.% with an accuracy of approximately ±10%, one can approximately estimate the relative concentration [H] from the expression [9]: $E_g$ = –0.9 + 0.09 × [H] (at.%, where [H] = H/(C + H)).

In this case, the values of the atomic ratio D/C and the color of the films, as shown by PL and XPS studies of the valence band [1,9], correlate as follows: in red and golden films D/C ≈ 0.5–0.8 (H/C ≈ 0.1–0.2) with the main PL band at 400–500 nm; D/C ≈ 0.2–0.4 in dark brown films (PL ~ 500–650 nm); D/C ~ 0.3 in brown–blue films. The maximum value of D/C ≈ 1.4 was obtained for translucent bright yellowish films [1] with a large value of $E_g$ ~ 3 eV, and the smallest one was obtained for brown–blue films with $E_g$ < 2.1 eV.

In this case, the photoluminescence energy (i.e., color), which is due to the fact of electron–hole recombination in $sp^2$ centers as a result of the excitation of electronic transitions from the valence band to the conduction band, is directly related to the value of $E_g$, when the PL wavelength increases with a decrease in $E_g$.

Thus, the information obtained from the XPS and XAES spectra of $CD_x$ films makes it possible to determine the following parameters:

(1) Elemental composition of the film; (2) relative concentrations of the elements and impurities, up to values of $\sim 10^{-3}$–$10^{-4}$ relative to carbon; (3) chemical state of the elements associated with the binding energy and the shape of the core lines; (4) the relationship of the $sp^2/sp^3$ states, according to the XPS C1$s$ and AXES spectra, i.e., graphite-like/diamond-like state ratio; (5) the valence band spectrum with an estimate of the $E_g$ value (related to the films' color) provides information on the metallic-semiconductor-dielectric nature of the films, as well as the presence of in-gap states, which may be impure or defective.

This is feasible for any material, including Be, Fe, W, and WC. This technique was also tested on mixed CH$_x$–Me films obtained using a QSPA-T machine [11]. Further, having determined the $E_g$ value from the valence band spectra, it is possible to approximately estimate the values of H/C from the above empirical ratio.

### 3.2.2. Relationship between the Color and Structure of Tungsten Erosion Products

For comparison, now consider the colors of tungsten films and W powder, which depend on the atomic ratio O/W, where W is a brilliant silver–gray powder, WO$_3$, one of the main products of tungsten erosion, is a lemon-yellow fine crystalline powder, and WO$_2$ is brown crystals, oxidizing to WO$_3$ when heated in air.

Tungsten hydroxides of a certain composition with an oxidation state of five to six are blue. Between the two valence-stable phases WO$_3$ (W$^{6+}$) and WO$_2$ (W$^{4+}$), tungsten forms several oxygen-defective oxides of variable composition (the so-called Magnelli phases) with a covalent-ionic type of chemical bond.

A typical defect of such Me 5$d$ materials is the deficiency in the oxygen sublattice. Theorists explain this by the fact that it is energetically more favorable to have a substoichiometric system with oxygen vacancies, i.e., WO$_{3-x}$ than WO$_3$ [21]. With vacancies, the WO$_{3-x}$ system acquires the properties of a semiconductor with $E_g < 3$ eV, and electron-type conductivity [22,23].

At the same time, surface oxygen vacancies, which are adsorption centers, determine the gas-sensing properties of thin films, including the ability to adsorb hydrogen isotopes, with a corresponding change in film conductivity (the principle of operation of chemoresistive gas sensors [23]).

The band gap values $E_g$ for WO$_3$ crystalline films vary from 2.6 to 3.25 eV depending on the type of optical transitions: direct or indirect. In addition, the $E_g$ value decreases with an increase in the fraction of oxygen vacancies, which provides an additional opportunity to characterize WO$_3$ using XPS, as in the case of CD$_x$ films.

When an injected electron enters an unfilled W5d orbital, a color center appears with an absorption maximum of approximately 1.3 eV in previously transparent WO$_3$ dielectric film. The removal of oxygen also means an electron doping, which leads to a coloring effect, as in the case of electron injection. In this case, the coloration associated with the appearance of the W$^{5+}$ charge state and a change in the optical properties can be obtained in several ways: electrochromism, photochromism, and thermochromism. In particular, with a variety of electrochromism from the action of the radiation at an energy $h\nu > E_g$, demonstrating the "coloring" effect due to the surface diffusion of the photon-induced charge of secondary electrons originating from the photoemission, e$^-$ + W$^{6+}$ (W$^{5+}$) → W$^{5+}$ (W$^{4+}$), one can detect a photon or soft X-ray beam with the help of WO$_{3-x}$ film screen [23].

Thus, heating to 500 °C leads to a gradual change in the oxides' composition, and the color of the film occurs: WO$_3$–WO$_{2.96}$ (green), WO$_{2.8-2.88}$ (blue), WO$_{2.7-2.75}$ (violet) and WO$_2$ (brown).

Electrochromism is associated with the existence of different final states screened by different numbers of W5d electrons. These valence states in the electronic structure of tungsten W = (Xe)4$f^{14}$5$d^4$6$s^2$ are W$^{6+}$, W$^{5+}$, and W$^{4+}$. In addition, during crystallization, the structure becomes not only ordered, but the film density also grows, which prevents electrochromism. In addition, the film heating changes the structure and color—, from amorphous and transparent to polycrystalline with a bluish tint; moreover, amorphous tungsten trioxide films have a certain ionic and electronic conductivity. As a result, using the

option of coloring films and powder in the system W plus oxides for the in situ observation and estimation of the O/W ratio is not such an easy task for tungsten wall erosion products, as in the above case of coloring $CD_x$ films formed during carbon wall erosion.

More precisely, coloring cannot be unambiguously applied for thin amorphous $WO_x$ films due to the possibility of the presence of the electrochromism effect (as well as thermochromism and photochromism) inside them but can be applied to polycrystalline structures formed at a higher temperature.

As confirmation for the case of thermochromism, in [24] a correlation was shown between the spectra of Raman light scattering (excited by a laser with $\lambda$ = 532 nm) and the color of tungsten trioxide $WO_3$ powder samples obtained by the thermal evaporation of a tungsten filament in a flow of gaseous argon or in a mixture of argon and water vapor.

The color of $WO_3$ can be related to both its stoichiometry and crystallinity. Thus, in sputtered thin films $WO_x$ ($2 \leq x \leq 3$), the transition from an $x = 2$ state starts in a dark blue color and ends at $x = 3$ with the formation of a transparent film (or white color for powders) after thermal treatment at 500 °C in an ambient atmosphere. After heat treatment (at 500 °C, 30 min) in an initially dark blue, bright blue, or sky blue color, the powders turn into crystalline white, as shown by Raman analyses for transition from an amorphous to crystalline $WO_3$ state, and this was also observed by the naked-eye.

The Raman spectra of $WO_3$ thermally treated powders of different colors were conventionally divided into three wavenumber zones [24], 45–450, 450–110, and 1100–1700 cm$^{-1}$, with each color having its own set of frequencies and peak intensities. For example, seven peaks of the low frequency Raman zone include navy-blue powders, while in the middle frequency, – zone navy-blue powders exhibit a band at 550–960 cm$^{-1}$, and the spectra of royal-blue powders include four bands near 79, 133, 263, and 323 cm$^{-1}$. At frequencies below 100 cm$^{-1}$, several peaks of different $WO_3$ phases (monoclinic, orthorhombic, and hexagonal) could be found, but the peak at 71 cm$^{-1}$ was characteristic of a monoclinic structure. The most intense O–W–O stretching modes for $WO_3$ were observed in the regions ~700–710 and 800–805 cm$^{-1}$.

Carbon contamination led to the appearance of peaks around 1440 and 1565 cm$^{-1}$, while the effect of hydrogenation resulted in a peak at 948 cm$^{-1}$. In addition, the surface oxygen vacancies of tungsten oxides can adsorb hydrogen.

As a result, an important practical result was obtained—; the color of polycrystalline $WO_3$ powders, observed with the naked eye, was compared with their structural properties, characterized by Raman spectroscopy, which plays an important role in the study of tungsten deposition, as is confirmed later.

### 3.3. Observation of the $W^{6+}$ Impurity in Mixed $CH_x$—Me Films from QSPA-T

In Figure 1 showing the XPS survey spectrum of the $CH_x$-Me film on a Si(100) substrate, clearly visible are tungsten peaks W4$f$, W5$s$, W4$d$, W4$p$, and W4$s$ (weak), since some of them have photoionization cross-sections $\sigma$ an order of magnitude higher than that of $\sigma$(C1$s$) = 0.013 Mb. Thus, $\sigma$(W4$f$) = 0.14 Mb, $\sigma$(W4$d$) = 0.22 Mb, and $\sigma$(W4$p$) = 0.1 Mb. As seen, the W peaks do not overlap with the main core lines of C1$s$ and O1$s$ (which are cut off by intensity), as well as with N1$s$ and with the impurity peaks of the erosion products Fe, Cu, Cr, and Si. In this case, the following elemental composition of the film and impurities (in at.%) was obtained: C1$s$ = 82.2, O1$s$ = 15.5, W4$f$ = 0.3, Si2$p$ = 1.1, N1$s$ = 0.4, Cu2$p_{3/2}$ = 0.2, Fe2$p_{3/2}$ = 0.2, and Cr2$p_{3/2}$ = 0.1.

In addition, all these impurity metals in $CH_x$-Me films exist in the form of oxides, since they interact in a vacuum chamber with residual oxygen thermally desorbed from the chamber walls under a powerful thermal load of ~0.5 MJ/m$^2$ on the W-target from the hydrogen plasma beam.

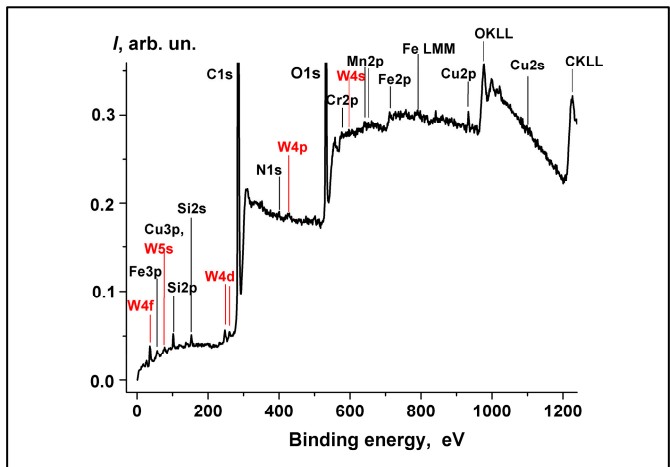

**Figure 1.** XPS survey spectrum of the CH$_x$–Me film on a Si(100) substrate, with the highlighted color of the tungsten peaks W4$f$, W5$s$, W4$d$, W4$p$, and W4$s$ (weak). The C1$s$ and O1$s$ peaks are cut off by intensity.

In connection with the above, and despite a small impurity fraction of W/C ~ 0.3 at. % in the XPS spectra and in the valence band (not shown) of the CH$_x$–Me mixed films [11], the W4$f$ doublet was clearly visible in Figure 2 at a binding energy of W4$f_{7/2}$ = 35.8 eV, corresponding to the WO$_3$ oxide.

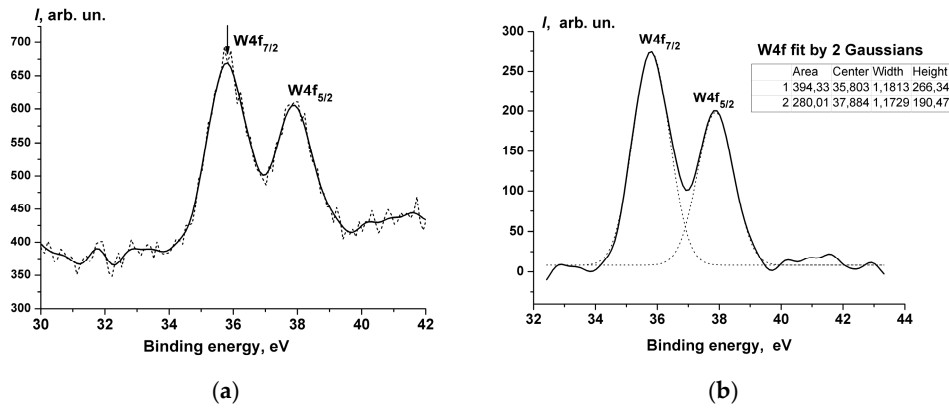

(**a**)                                          (**b**)

**Figure 2.** (**a**) XPS spectrum of CH$_x$-Me film in the region of the W4$f$ doublet; (**b**) W4$f$ doublet fitting by 2 Gaussians.

The XPS measurements in the region of the W4$f$ doublet (Figure 2a,b) show the peak positions W4$f_{7/2}$ = 35.8 eV, and W4$f_{7/2}$ = 37.9 eV, as features of a tungsten trioxide WO$_3$ with a W$^{6+}$ valence state. This also corresponds to the known presence of two stable phases during the erosion of tungsten, WO$_3$ (W$^{6+}$) and WO$_2$ (W$^{4+}$). With the temperature growth during oxidation, a slight shift in the position of W4$f_{7/2}$ is observed up to approximately 0.5 eV towards lower binding energies. This is due to the fact of a small partial decrease in the degree of tungsten oxidation from W$^{6+}$ to W$^{5+}$, i.e., with the formation of the above oxygen vacancies upon heating [25].

The peaks intensity ratio in the W4$f_{7/2,5/2}$ doublet is proportional to the degree of the degeneracy of the 2 $I$ +1 state, which for the $f$—state with the orbital momentum $L = 3$ and the total electron momentum $I = L \pm S$ (spin $S = 1/2$) results in the "standard" intensity ratio of 4/3 and S-L spin-orbit splitting of 2.1 eV.

In addition, in the spectrum in Figure 2a, there is no phase of the tungsten metal doublet W(0): W4$f_{7/2}$ = 31.0 eV and W4$f_{5/2}$ = 33.2 eV, and there is also no phase of tungsten dioxide WO$_2$ for W4$f_{7/2}$ = 32.4 eV and W4$f_{5/2}$ = 35.5 eV. There are also no traces of the WC carbide phase in Figure 2, for which its peak, W4$f_{7/2}$, should be in the binding energy

range of 31.7–31.8 eV, relative to the Fermi level, according to the XPS data from the spectra of WC nanopowder with $WO_3$ impurity (Figures 3 and 6 in [26]). However, in the presence of all these phases, they would be well fixed even against the prevailing accumulation of carbon component.

As known, in tokamaks, the erosion products of W (an element with high valence) in the form of tungsten films and tungsten dust are oxidized, especially since the vacuum is not ideal, and as the temperature rises, oxidation intensifies, including as a result of oxygen desorption from the walls of the chamber (after adsorption of atmospheric gases) under the influence of plasma flows, and oxidized dust usually remains inside for a long time [27], though small dust particles may be more efficiently destroyed during discharges.

Thus, the most stable tungsten erosion product is often observed as $WO_3$. It was also shown that all impurity metals, such as W4$f$, Cu2$p$, Fe2$p$, and Cr2$p$ in $CH_x$–Me films from the QSPA-T interact in a vacuum chamber with residual oxygen thermally desorbed from the chamber walls under a powerful thermal load of ~0.5 $MJ/m^2$ on the W-target from a hydrogen plasma beam.

## 4. World Systems for In Situ Analysis of Erosion Products of First Wall Materials

The practical implementation of the MAPP system (materials analysis particle probe) for the in situ analysis of the erosion products of first wall materials during tokamak operation was carried out on the NSTX-U tokamak with the support of the Princeton Plasma Physics Laboratory (PPPL, Princeton, NJ, USA), Purdue University (USA) and the US Department of Energy [28,29]. The challenge was to provide a fast and direct analysis of plasma-facing components exposed to NSTX plasma discharges within a minimum time window of ~12 min between plasma shots, and several samples could be inserted simultaneously at the level of the plasma-facing wall surface.

The indicated introduction of samples turned out to be close to the mechanism of introducing into vacuum chamber the Si(100) film substrates on rods to the level of the wall surface, which was conducted earlier on the T-10 tokamak, when studying the electronic structure of $CD_x$ films depending on the controlled deposition conditions [9]. In this case, only the working plasma or cleaning discharges affected on the Si substrate, followed by the removal of irradiated samples for ex situ analysis using XPS.

With the MAPP probe, observations were made in situ using such methods as XPS, ion scattering, direct recoil spectroscopy (DRS), and thermal desorption spectroscopy with a quadrupole mass spectrometer, performed immediately after the end of plasma discharges, i.e., without interrupting the NSTX-U operation. The XPS and TDS techniques were also used in experiments with $CD_x$ films. Let us briefly consider some elements of the MAPP system.

The vacuum in the MAPP chamber is no worse than $5 \times 10^{-7}$ mbar, and the chamber itself is isolated from the volume of the NSTX tokamak by two automatic vacuum valves connected in series. Four test specimens on rods are attached to the head of the MAPP probe and moved at the level of the divertor for irradiation with plasma discharges.

After the terminal of the magnetic field pulse of ~1 T and its rapid decay, ambient fields of ~10 G remain. The electron and ion analyzer in the MAPP (i.e., Comstock hemispherical analyzer) is operated in laboratory magnetic fields of 16 G without interference from the fields on the obtained spectra. For shielding from low-frequency magnetic fields, the analyzer is housed inside a high-permeability mu-metal metal case. The analyzed electrons and ions are recorded by multichannel plates in a chevron configuration.

The chemical surface composition is studied using the XPS with Mg ($h\nu$ = 1253.6 eV) and Al (1486.6 eV) anodes at a probing depth of ~8 nm, and the upper 1–2 monolayers were probed using ion scattering spectroscopy with a $He^+$ source of ~1.5–3 keV energy.

Direct recoil spectroscopy is a variation of ion scattering spectroscopy with the same NTI 1404 ion source but with a different $Ne^+$ gas. Heavy ion projectiles of the $Ne^+$ type have a much higher scattering cross-section for recoil atoms than light ions of the $He^+$ type. The recoil ions are detected by the analyzer. This makes the DRS method one of the

few methods that can directly detect atomic H on a surface using direct light ion recoil spectroscopy. The energies of scattered and reflected ions in the MAPP are analyzed almost in the same way as electrons in an XPS spectrometer, with one single analyzer, since by scanning the voltages on a hemispherical analyzer, one can obtain the energy spectrum of the detected ions.

For the TDS measurements, a temperature ramp (~1 °C/s) is supported with the registration of desorbed ions by the Inficon 100L RGA quadrupole mass spectrometer (Inficon Co., Bad Ragaz, Switzerland).

MAPP is used along with an ex situ tile analysis, experiments on laboratory facilities, accelerators, and model calculations.

Another example of in situ studies of erosion products is the PRIHSM (particle and radiation interaction with hard and soft matter) plasma facility at Purdue University (USA, Indiana), designed to study the surface of samples in ultrahigh vacuum, which makes it possible to modify the surface with an ion beam and characterize it in situ using XPS, ultraviolet photoelectron spectroscopy, angle-resolved photoemission spectroscopy, and low-energy ion scattering spectroscopy using a $He^+$ ion beam at 1500 eV [30].

A variation of the MAPP is also the IGNIS (ion-gas and neutrals interactions with surfaces) plasma facility for the interaction of ions, gas molecules, and neutrals with a surface for surface modification and defining its characteristics in situ at the Illinois University (USA) [31].

In addition, in situ and ex situ studies of metal (Mo, W, and Ta) and carbon erosion products were carried out on the TEXTOR limiter tokamak using in situ spectroscopy elements: a spectrometric CCD video camera and an infrared CCD video camera with interference filters [32].

The subsequent ex situ analysis was carried out by various methods of ion-beam analysis as the most effective, associated with irradiation at accelerators by a monochromatic collimated ion beam with light ions $H^+$, $^3He^+$, and $^4He^+$ with energies ~MeV (using methods for analyzing nuclear reactions) and energies ~keV (in the method of ion etching with mass spectrometric analysis). The analysis of the reaction products, including laterally and in depth, was carried out using nuclear reactions, for example, such as $^{12}C(^3He, p)^{14}N$, $^{13}C(^3He, p)^{15}N$, $d(^3He, p)\alpha$, and $^9Be(^3He, p)^{11}B$, as well as SIMS (secondary ion mass spectrometry). In addition, radioactive methane $^{13}CH_4$ was used as a marker for ex situ studies of carbon migration along divertor tiles.

Thus, the used method of ion-beam analysis with the participation of light ions with energies ~MeV at accelerators and radioactive isotopes makes it possible to noticeably advance into the bulk of the studied erosion products up to a ~10 μm depth and more.

With an in situ monitoring system, it is possible to constantly control the process of film growth under a plasma flow within the intervals between plasma pulses at practically any sample thickness.

Finally, in [33], the importance of developing in situ methods is emphasized, especially in linear plasma installations for studying the erosion of plasma-facing materials at a high energy flux density. However, despite their advantages, they have a significant drawback, which is high complexity and cost. Therefore, most linear plasma experiments still rely on ex situ surface analysis techniques. Another reason for the preferential use ex situ, is that samples can be transported to different laboratories specializing in specific methods of analysis, which increases the versatility and quality of analyzes.

In addition, as additional examples of in situ analysis, the DIONISOS (MIT, Cambridge, MA, USA) and PISCES-B (Japan) plasma devices are presented, where the mentioned techniques such as XPS, XAES, SIMS and laser techniques LIDS (laser-induced desorption spectroscopy) and LIBS (laser-induced breakdown spectroscopy), are available. Note that LIBS spectroscopy has been considered in recent years as one of the most promising methods for the quantitative determination of hydrogen isotope (H, D, and T) content in plasma-facing components of thermonuclear devices in JET and ITER [34].

Within the framework of the EUROfusion consortium program, work was carried out taking into account the following basic requirements for ITER: quantitative analysis of the fuel from the corresponding surfaces; technical demonstration of the LIBS implementation with remote control system; accurate detection of fuel at atmospheric pressure, which is especially important for ITER. For the first purpose, the elemental composition of the ITER-like deposits, including deuterium (D) or helium (He), contained in erosion products of the W–Be, W, W–Al, and Be–O–C types was successfully determined with a typical resolution in depth from 50 to 250 nm per laser pulse. Deuterium was used as a substitute for tritium (T), and in the LIBS experiments its surface density $< 10^{16}$ D/cm$^2$ was measured with an accuracy of ~30%, which confirms the required high sensitivity of the method.

For the second purpose, the operation of LIBS with remote control inside the Frascati-Tokamak-Upgrade tokamak was demonstrated, where the compact LIBS system was installed on a manipulator introduced into a vacuum volume for analysis within the intervals between plasma discharges. Concerning the capabilities of LIBS under pressure conditions relevant for ITER, at pressures up to 100 mbar N$_2$, the D content was determined with an accuracy of up to 25%, and at atmospheric pressure the accuracy was approximately 50%. In this case, the W, Al, H, and D emission lines (including the H$_\alpha$ and D$_\alpha$ lines of the Balmer series with $\lambda \sim 650$ nm) were clearly recorded in the plasma plume in the mode of a single laser pulse.

## 5. Monitoring the Accumulation of Hydrogen Isotopes in Carbon Films Using IR and Raman Spectroscopy

### 5.1. IR Vibrational Modes of CD$_x$ Films

The IR reflectance spectra of CD$_x$ flakes were obtained using a Bio-Rad UMA500 microscope with 500× magnification attached to the frame of a Bio-Rad FT-IR FTS3500 Fourier spectrometer operating in the reflection mode in the IR range of 7000–400 cm$^{-1}$ (with a sensitivity decrease at 700–400 cm$^{-1}$), which covers the main vibrational modes of the C-, D-, H-, and O-components of the films and the carbon net, and with a spatial resolution of <8 μm. The measurements were carried out in the reflection mode, with a low reflection coefficient and a structureless shape in some regions of the reflection spectrum, indicating strong absorption, i.e., as if saturation of the vibrational modes due to their overlap and strong absorption.

As an example, Figure 3 in the frequency range 4000–400 cm$^{-1}$ shows the IR reflection spectrum of the "red" (in color) CD$_x$ flake No. 2 with a thickness of 30 μm, for which the ratios D/C = 0.57 and H/C = 0.23, typical for such films, were measured by nuclear physics methods (see paragraph 2). Film No. 2 appears to be slightly heated and has a more uneven surface, according to a lower reflectance of 0.02–0.06.

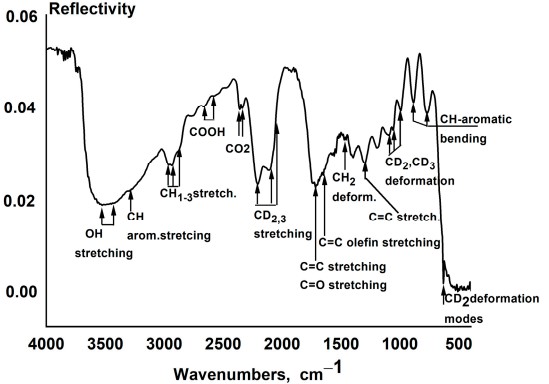

**Figure 3.** IR reflectance spectrum of CD$_x$ flake No. 2, 30 μm thick, with D/C = 0.57 and H/C = 0.23, at room temperature, with the indication of the main vibrational modes and their band positions.

In Figure 3, the range of low-energy modes 700–1100 cm$^{-1}$ can be attributed mainly to the following modes [35]: bending vibrations of CH$_n$ groups in the aromatic ring outside the plane, 793 cm$^{-1}$, 877 cm$^{-1}$, and in the plane 968 cm$^{-1}$, as well as skeleton stretching modes C–C 1054 and 1135 cm$^{-1}$. In addition to the CD$_2$ $sp^3$ deformation modes (bending or rocking) at 1092 and 633 cm$^{-1}$, the CD$_3$ $sp^3$ deformation modes at 1056 and 992 cm$^{-1}$ are also noticeable [36]. The most intense are the stretching modes (i.e., valence vibrations) $sp^3$ CD$_{2,3}$ at 2300–2100 cm$^{-1}$, which are a "passport" of carbon–deuterium erosion products in the form of CD$_x$ films. These include modes at 2073, 2117, and 2217 cm$^{-1}$ [9].

The peaks at 2341 and 2364 cm$^{-1}$ are related to the CO$_2$ modes. Their intensity is determined mainly by the constancy of the atmosphere during the measurement. In the region of 2900–3100 cm$^{-1}$, there are characteristic $sp^3$ CH$_n$ ($n$ = 1–3) hydrogen stretching modes [37] of hydrocarbon erosion products. The presence of CH $sp^2$ olefin modes at 3020, 3000, and 2950 cm$^{-1}$ in linear C=C bonds is hardly noticeable, contrary to those of CD modes. Features at 3200–3600 cm$^{-1}$ belong to the stretching modes of O–H hydroxyls [37].

A specific feature of the IR spectra of CD$_x$ films is the presence of broad lines ranging from a full width at half maximum of 15 cm$^{-1}$ (CO$_2$ modes) to 40–60 cm$^{-1}$. According to [38], the reason for this may be the strong influence of the hydrogen bond. In this case, it may be due to the fact of a high concentration of C–H groups and the presence of O–H groups. This is most noticeable for the O–H stretching modes at 3200–3600 cm$^{-1}$. Hydroxyl groups do not exist in isolation, but are linked to other O–H groups through an extensive hydrogen bond. Apparently, this also applies to the effect of hydrogen bonding on the C–H, C–O, and possibly C–D groups via O–H hydrogen bonds between the O and H elements of neighboring polymer chains in the film structure. Such cross-linking in macromolecules, which is known for polymers, as well as the strong $sp^3$ skeleton of the C–H and C–D modes, cause well known high residual mechanical stresses and high brittleness of smooth CD$_x$ films (leading to dust formation).

Moreover, for CD$_x$ films, the in situ control was found empirically over the adsorption and desorption of hydrogen isotopes in films and on diagnostic mirrors by IR vibrational modes depending on the deuterium atomic concentration D/C and film thickness. Thus, at D/C $\approx$ 0.2, it was observed only in the $sp^3$ CD$_2$ deformation modes at wave numbers of 633 and 1090 cm$^{-1}$ for a film thickness of at least 0.06 μm. Then, with a concentration growth at D/C $\approx$ 0.3–0.6, it became observable in $sp^3$ CD$_{2,3}$ stretching modes at 2100–2200 cm$^{-1}$ at a thickness of at least 0.3 μm [39].

When comparing studies of CD$_x$ films by IR and Raman methods, we must note the following practical remarks. According to the differences in the light absorption coefficient in CD$_x$ films in the near IR region with a wavelength of λ ~ 6 μm (~$10^3$ cm$^{-1}$), and during Raman studies (by a laser excitation in the optical range of ~500–600 nm), a higher absorption coefficient, ~$10^4$ cm$^{-1}$, is present [40]. As a result, according to our practical observations, more efficient detection of hydrogen isotopes by the most noticeable stretching IR sp$^3$ CD$_2$ and CD$_{2,3}$ modes begins for CD$_x$ film thicknesses above ~0.1–1 μm, and up to a thickness of ~10 μm [39], while the Raman modes may be detected at a noticeably less film thickness: ~10–100 nm [41].

However, in the Raman method, the strong influence of the fluorescence background in CD$_x$ films, due to the recombination of electron-hole pairs in $sp^2$ clusters as luminescence centers, suppresses the Raman signal at a high relative content of hydrogen isotopes H(D)/C > 0.4 [42]. Let us consider these observations in more detail.

*5.2. Raman Spectra of CD$_x$ Flakes and a-C:H:D Thin Films*

The first Raman experiments with CD$_x$ gold flakes (D/C = 0.5–0.8) using laser excitation with λ = 1064 and 514 nm did not allow registration of Raman spectra due to the high fluorescence background. This was conducted only with dark brown flakes. However, the presence of hydrogen isotopes with H/C > 0.4 can also be indirectly detected, but only by a significant growth of the background intensity and its shift to a maximum position of ~3000–5000 cm$^{-1}$, as the H/C ratio grows.

The Raman spectra of amorphous carbon films are more "smeared" [42] than those of crystalline ones (such as $WO_3$ samples [24]) due to the disordered structure. All carbon structures have common features in the Raman spectra in the region of ~800–2000 $cm^{-1}$. This includes the so-called main G (graphite) and D (disorder) peaks located in the region of ~1560 and ~1360 $cm^{-1}$, respectively, as well as the T peak in the region of ~1060 $cm^{-1}$ [42]. The G peak is due to the stretching in-plane vibrations with $E_{2g}$ symmetry of all pairs of $Csp^2$ atoms in aromatic rings and in linear olefin chains (–C=C–). The D peak is due to the "breathing" modes with the symmetry $A_{1g}$ of $Csp^2$ atoms in aromatic rings, and it indicates the appearance of disordering, with the growth of which the D peak intensity the also increases.

Dark brown $CD_x$ films were measured using a LabRAM HR Visible spectrometer (Horiba Jobin Yvon) equipped with a microscope with a lateral resolution of 1 μm, a TV camera and a cooled CCD detector, excited by a He-Ne laser with λ = 633 nm and a maximum power of 20 mW, with spectrograph resolution <1 $cm^{-1}$. In this case, the laser beam is focused on the sample over an area of several $μm^2$, penetrating to a depth of more than 100 nm. Among the 10 studied areas on different flakes, only 8 succeeded in showing a weak Raman signal with a signal-to-noise ratio = 10–12, and having a slope of the background curves $m = \Delta y/\Delta x \approx$ 4–10 cm, where $\Delta x$ scan is measured in units of $cm^{-1}$, and $\Delta y = I(G)$ peak intensity.

Figure 4a shows two of the most typical of the eight measured Raman spectra for samples T3 and T5 with peaks D at 1366 and G at 1571 $cm^{-1}$, normalized to the intensity maximum for comparison, and Figure 4b,c show their decomposition into typical Gaussians lineshapes.

**Table 2.** D and G peak parameters for T3 and T5 films.

| Sample | Peak D, $cm^{-1}$ | | Peak G, $cm^{-1}$ | | I(D)/I(G) |
|--------|-----|------|-----|------|-----------|
|        | ν   | FWHM | ν   | FWHM |           |
| T3     | 1370 | 152 | 1571 | 117 | 0.75 |
| T5     | 1378 | 113 | 1574 | 95  | 0.94 |

As seen in Figure 4b,c, a common element of the spectra is the presence of a dip between the D and G peaks at approximately 1500 $cm^{-1}$, which is a typical feature of a-C:H polymer films [43]. This band close to 1500 $cm^{-1}$ position is represented by a Gaussian at T3 and T5, which arises from amorphous carbon with $sp^2$ bonds, or fragments and functional groups in a disordered structure [44]. As seen in Figure 4a, the T5 sample has an additional shoulder at about 1208 $cm^{-1}$ (and T3 has a sloping shoulder), and a low-intensity shoulder at approximately 1060 $cm^{-1}$ (arrow), indicating the presence of a weak T-peak, which makes the T5 spectrum slightly similar in shape to that of the technical soot [44].

In addition, the presence of the T peak is due to a more noticeable disorder for the T5 sample, which also manifests itself in an increase in the $I(D)/I(G)$ intensity ratio, seen in Table 2. The first decomposition bands of the spectra T3 (Figure 4b) and T5 (Figure 4c) in the region of singularities near 1200 $cm^{-1}$ with half-widths of 113 and 152 $cm^{-1}$ are assigned to $sp^2$–$sp^3$ bonds or C–C and C=C stretching vibrations of trans-polyacetylene-like structures, according to [44]. Further, based on the approximate empirical dependence of the relative deuterium content H(at.%) = H/(C + H) (at H = 20–50 at.%) versus slope of the spectrum $m$ (measured in microns) and intensity signal $I(G)$:

H(Raman) = 30.0 + 4.6 × lg($m/I(G)$ μm) (at.%) [45], we get the following. With a slope $m$ = 4.65 cm for the T5 spectrum and $m$ = 9.64 cm for T3 (with a 15% higher signal intensity and a higher fluorescence background), we obtain the value of the relative concentration of deuterium in the films: H(T5) = 38.0 at.% and H(T3) = 39.2 at. %. This is close to the Raman detection limit depending on the high background of the fluorescence at high ratios of H(D)/C > 0.4.

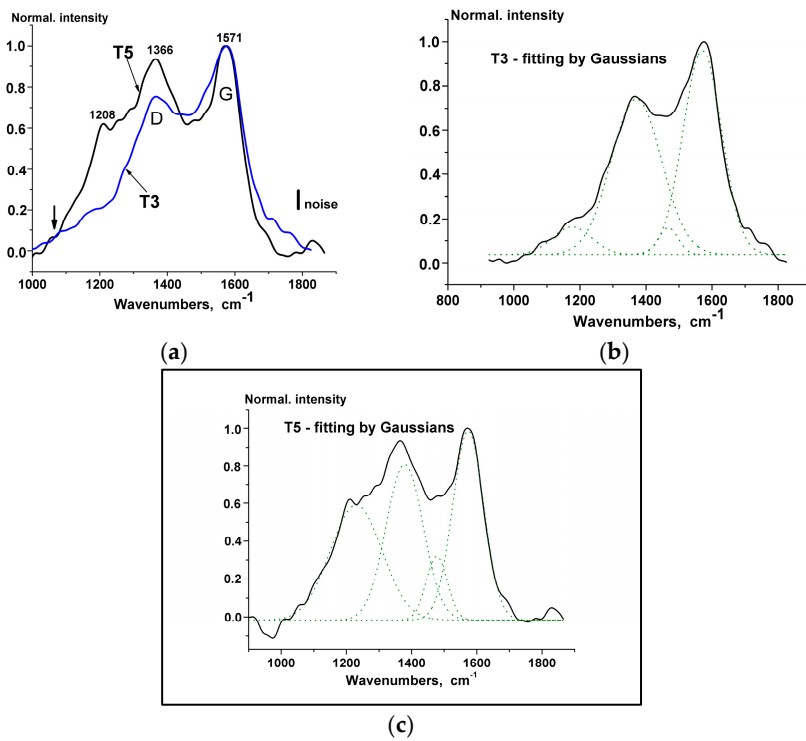

**Figure 4.** (**a**) Typical Raman spectra of $CD_x$ films T3 and T5, normalized to the maximum intensity. The arrow is approximately 1060 cm$^{-1}$; (**b**) Raman spectrum of the T3 film normalized to the maximum intensity and its decomposition into 4 Gaussians (dotted lines at 1176, 1371, 1468, and 1571 cm$^{-1}$), see Table 2; (**c**) Raman spectrum of the $CD_x$ T5 film normalized to the maximum intensity (solid line) and its decomposition into 4 Gaussians (dotted lines, at 1229, 1378, 1477, and 1574 cm$^{-1}$).

However, these two values are below the Raman detection limit for the background growth of photoluminescence at H = 45–50 at.%. These values are also lower than H ≈ 50–60 at.% for $CD_x$ gold films.

Thus, in Raman studies of carbon films with large values of H(D)/C > 0.4, despite the absence of an observed signal with the T, D, and G modes under a strong fluorescence background in the region of ~1000–2000 cm$^{-1}$, the very existence of such a background indicates the presence of high-concentration hydrogen isotopes. This can also serve as an indicator of hydrogen accumulation, which is accompanied by the rapid growth of the background intensity and its maximum shift to the region of higher frequencies ~3000–5000 cm$^{-1}$. As will be shown below, the fundamental $WO_3$ modes are located in the lower frequency range 1453–400 cm$^{-1}$.

The temperature dependence of the deposition of thin a-C:H:D films 20–300 nm thick on stainless-steel (SS316) and on molybdenum metal mirrors deposited in $Ar/D_2$ magnetron discharge plasma with a graphite cathode, was studied using Raman and ellipsometry methods [41]. With the temperature growth, the Raman spectra revealed a significant intensity decrease in the T peak (~1060 cm$^{-1}$) from the $sp^3$ states, a decrease in the $I(D)/I(G)$ ratio, as well as smeared D and G bands (~1360 and ~1560 cm$^{-1}$), which indicate a strong amorphization of the films with temperature growth at $T > 500$ K and the predominance of $sp^2$ bonds. In addition, the temperature of the "zero growth rate" of the film was found around 500 K, which can be recommended in the ITER optical diagnostics for protecting metal mirrors from parasitic hydrocarbon deposition during heating.

## 6. Raman Spectroscopy of Carbon Dust in the NSTX Tokamak

It is interesting to note, that an example of the effective use of Raman spectroscopy is the detection of carbon dust in the NSTX tokamak with plasma materials made of graphite and tiles from a carbon fiber composite. This method has been successfully used to control

carbon dust formed from plasma-irradiated materials in NSTX plasma discharges, as well as in comparison with the impact on dust of an arc discharge in an argon atmosphere, and when heated in vacuum to 2700 K [46].

Thus, under plasma irradiation, the intensity of the D-peak at 1350 cm$^{-1}$, compared with the G-peak at 1580 cm$^{-1}$, increased due to the transition from the state of ideal graphite, such as annealed pyrolytic graphite, through the industrial graphite phase towards highly disordered (amorphous) black carbon type graphite. This growth is explained by an increase in the amount of disordered carbon and a decrease in the size of graphite crystals, since it is known from X-ray diffraction analysis that the ratio of the intensities of the D-mode to the G-mode is inversely proportional to the crystal size.

It is interesting that the shapes of the D and G lines of dust differed from each other in the case of an arc discharge, and when heated in vacuum up to 2700 K. The latter circumstance indicates the efficiency of carbon dust detection by the Raman method, and the unique possibility of recognizing even different types of carbon dust from the spectrum.

For comparison, as shown in Figure 4a, the Raman spectrum of dark brown $CD_x$ flakes from T-10 with the presence of the $sp^2 + sp^3$ modes of trans-polyacetylene and amorphous carbon has a more complex shape and a larger fraction of deuterium $x \sim 0.38$ in $CD_x$ films than that of the indicated carbon dust. This also confirms the sensitivity of the Raman method to various types of plasma action on carbon materials, and this method is used effectively to characterize tungsten-containing materials, as shown below.

## 7. Study of Divertor Carbon and Tungsten Materials from the JET Tokamak

In [47], the study of JET erosion products obtained with different wall materials and in two different campaigns is reported: JET-C (with carbon fiber composite) and JET-W ("ITER-like" walls with tungsten materials), irradiated with hydrogen and tritium. There were used techniques, such as FTIR, Raman, TDS, EPR, TGA, and EDX for the impurities. Let us consider the main results of these studies and compare the results for JET carbon materials with those for $CD_x$ films from the T-10 tokamak.

The obtained FTIR signals of the erosion products of tungsten walls in the frequency range 400–1000 cm$^{-1}$ are caused by metal–oxygen bonds W–O and Fe–O, where the W–O modes occupy the range of 805–810 cm$^{-1}$. These modes can serve as a monitor for the accumulation of tungsten erosion products in the form of W–O modes due to the inevitable presence of residual oxygen impurities, while H atoms, –OH groups, and $C_xH_y$ hydrocarbons can be adsorbed on oxygen vacancies in the modes of W–O oxides.

Recall, that in the case of the T-10 tokamak, the accumulation of hydrogen isotopes in $CD_x$ films was recorded using the $CD_{2,3}$ IR modes at approximately 2100–2200 cm$^{-1}$, etc., as well as the above Raman modes.

The IR spectra of irradiated carbon tiles showed a difference from non-irradiated tiles in the region of 400–900 cm$^{-1}$ due to the accumulation of hydrogen isotopes [47]. However, the authors did not identify the infrared modes of hydrogen isotopes of the C–H, C–D, or C–T type in the bands at 490, 570, 650, 710, and 820 cm$^{-1}$. The authors attribute the IR spectra of gases with bands at approximately 670 and 2300 cm$^{-1}$ obtained by thermal desorption to $CO_2$ desorption, weak peaks at 1400 and above 3000 cm$^{-1}$—to the O–H modes, and the intensity of the O–D modes turned out to be below the registration limit, i.e., hydrogen modes were also found in traces of $H_2O$ and $D_2O$ vapors.

For comparison, using IR spectroscopy, the study of carbon erosion products such as free standing $CD_x$ films turned out to be more sensitive than a similar study of erosion products in the form of films adsorbed on thick JET-C tiles. In addition, the detection of the W–O modes in the erosion products of tungsten, formed as a result of the presence of residual oxygen, turned out to be easier to identify than the modes of hydrocarbon isotopes, according to the authors.

The EDX analysis showed the presence of metallic impurities (Fe, Ni, W, and Be) and non-metallic N and O, as well as carbon. In the Raman spectra of the erosion products,– (i.e., flakes and dust), D and G peaks at 1350 and 1580 cm$^{-1}$ were observable on sample

pieces cut from carbon wall tiles. In this case, in Raman studies, as in the indicated case of $CD_x$ films, an increased luminescence background was observed, with an increase in the fraction of the hydrogen isotopes content $x > 0.4$ due to the fact of carbon impurities. The Raman peak from tungsten oxide was observed at approximately 810 cm$^{-1}$.

As expected from detailed measurements of thermal desorption for samples exposed with tritium, the tritium content for the tungsten erosion products turned out to be 10–20 times lower than that for carbon products. Thus, the authors have shown the possibility of registering the accumulation of metal and carbon deposits using IR, Raman, and TDS spectroscopic methods.

The EPR measurements (9.8 GHz, 200 and 1000 Gauss H-field scanning) showed two main paramagnetic centers with the Lande factors $g = 2.002$ and $g = 2.12$. The signal with $g = 2.002$ can be associated with paramagnetic centers in the aromatic compounds with delocalized π-electrons from multi-ring aromatic structures.

EPR spectroscopy offers information about the density of unpaired electron spins of carbon atoms, the shape of the spectral line, and its width [20]. The shape of the EPR line is determined by the type of interaction between unpaired spins and the environment, including the paramagnetic complexes of $3d$ metal impurities (Fe, Cu, Cr, Ni, etc.), having an unpaired spin. The line width depends on the strength of the interaction, and the value of the g-factor correlates with the size of the band gap and the fraction of the sp$^2$ states.

The value $g = 2.00220$ was obtained in our case for a-C:H films with a high ratio of $sp^2$ states, which is close in value to carbon deposits in the T-10 tokamak [20]. However, this differs from $CD_x$ films with larger value of $g = 2.00341$, and $sp^3 \sim 0.7$. The EPR signal with $g = 2.12$, according to the authors, is not associated with carbon structures, but may be due to the impurities of nickel ferrites, as well as to the presence of other ferrites and oxides. These results turned out to be qualitatively close to those obtained in [20] for different types of $CD_x$ flakes deposited as a result of the erosion of the MPG-8 graphite limiter in the T-10 (in contrast to carbon fibers in the JET-C divertor) when the EPR data were also obtained (9.9 GHz, scanning with a field of 100 and 6000 G), including those with growing temperature. In addition, an analysis of the lineshape and width of the EPR lines was conducted, which is not available in the cited work. For the so-called "red" in the color of the $CD_x$ films with $x \sim 0.7$ (and $sp^3 \sim 70$ at.%), it showed a value of $g = 2.00341$ and a Lorentzian lineshape; while the "dark" films with $x \sim 0.4$, $g = 2.0029$ showed a lineshape in the form of a distribution of quasi-Lorentzians.

In addition, the major share of the paramagnetic impurities of the $3d$-metals recorded using the EPR method (which, important to note, affects the bulk properties of the considered thickness of ~20–30 μm of $CD_x$ films), originating from the erosion of the chamber walls, was iron $Fe^{3+}$ (~0.7 at.%) with spin $S = 5/2$ and $g$ factors (in descending order of signal intensity) equal to ~2.07 (broad line), 4.3, and 9.9 [20]. They correspond to different positions of the $Fe^{3+}$ ion impurity with spin $S = 5/2$ in the carbon network of the film. Thus, the broad line with $g \approx 2.07$ refers to $Fe^{3+}$ ions predominantly in the Fe–C state; however, the presence of Fe–O and Fe–OH bonds in a more symmetrical octahedral structure in the bulk of the film is also possible in the presence of a weak crystal field.

Thus, a comparison of the erosion products of the JET and T-10 showed the efficiency of recording the accumulation of both metal and carbon impurities using IR, Raman, and EPR spectroscopies, and also it revealed differences in the properties of the carbon films in the JET and T-10 formed due to the differences in the carbon wall materials and their plasma sputtering conditions, as well as the advantage of studying carbon erosion products in the form of free-standing films of the $CD_x$ type compared to films on the surface of thick tiles.

## 8. Examples of the Tungsten Materials Studied by IR Spectroscopy, XPS and Other Methods

In [48], dust-free W-substrates were exposed to deuterium plasma from a GyM linear plasma machine (Institute of Plasma Physics, Milan) at an ion fluence of $\simeq 2.9 \times 10^{24}$ m$^{-2}$

with a kinetic energy of $E_i \sim 400$ eV and an HF argon plasma glow discharge with an energy $E_i \sim 200$ eV. In this case, the chemical composition of the surface changed due to the fact of physical sputtering. The aim of this work was to quantify the effect of typical atmospheric pollutants on W-on-W adhesion, which are modeled by exposure to these plasma beams since the surface changes under irradiation.

Thus, before plasma exposure, the IR spectrum was dominated by vibrational bands attributed to the OH groups such as W–OH at approximately 1100 and 1700 cm$^{-1}$, whereas, after exposure, the oxide bands W–O–H, W–O at approximately 2000–2200 cm$^{-1}$ began to predominate in the spectrum, and the OH bands almost disappeared. In addition, the measurement of the adhesion of the W-on-W conductive dust of micron sizes (which is determined by the interactions between the induced multipoles and not by a metallic bond) in the presence of electric fields showed it decreased by 50% as a result of the modification of the chemical composition surfaces, leading to the formation of tungsten oxides.

Note that in the case of carbon materials, the presence of nonconductive dust and dielectric films on the first wall, the current–voltage characteristics of which were obtained in [1,49] for $CD_x$ films from the T-10 tokamak, can lead to the accumulation of an induced charge from charged particles in plasma, either from a secondary emission or thermionic emission, according to [50]. This may contribute to the edge localized mode (ELM) events in tokamaks or lead to unipolar arcing due to the strong field electron emission and indicates the importance of controlling the conductivity of the erosion products, which was noted earlier during the current–voltage study of $CD_x$ films.

Indeed, in [49] for "red"-colored $CD_x$ films with a fraction of diamond-like states $sp^3 \approx 70$ at.%, it was shown that on the plasma side of the film, "looking" into an electric field, high-resistance charge states start to form. They show the low mobility of the charge carriers and presence of charge traps with a long lifetime after the electric field is off (tens of minutes). This is well known for amorphous materials such as wide-gap a-C:H films with a resistivity $\rho \sim 10^8$–$10^9$ Ohm $\times$ cm, showing the same features.

As a result, plasma-induced charges can accumulate on such a film in a tokamak. With a growth of the $sp^2$ ratio, the $\rho$ value decreases, while for the brown or dark-brown $CD_x$ films, this degradation reaches 2–4 orders of magnitude (which will noticeably accelerate the discharge of the traps for a given "threshold" value of $\rho$), and for graphite with $sp^2 \approx 100$ at.%, the value of $\rho$ becomes close to zero. In this case, with an increase in the fraction of the $sp^2$ states, the band gap $E_g$ decreases, which, like the fraction of the $sp^2$ states, can be found from the XPS C1$s$ spectra, XAES of the CKVV Auger electrons, and the $E_g$ value can be estimated from the valence band spectrum, as stated above.

If necessary, for a more detailed analysis, it is possible to carry out calibration experiments to determine the dependence of $\rho$ versus the $sp^2$ ratio, and $\rho$ versus $E_g$, —from the XPS C1s valence band spectra, and estimate the "threshold" value of $\rho$ for the charging effect.

The quantitative in situ control of the film conductivity and the amount of dust in fusion devices can be conducted by measuring the surface charge using a non-contact capacitive method [51,52].

As a result, we note that with the accumulation of carbon dust (as well as adsorbed hydrogen isotopes), it is possible to identify and control their safe level by the intensity growth of the corresponding signals in three ways: Raman spectrum (see Figure 4a–c), indicated IR modes $sp^3$ $CD_{2,3}$ at approximately 2200 cm$^{-1}$, and C–C carbon net modes at 1054 and 1135 cm$^{-1}$ (Figure 3). At the same time, using the C1s XPS spectrum, it is possible to estimate the ratio of $sp^2$ graphite-like states, which growth contributes to the increase of conductivity, and, for example, it is also possible to carry out calibration experiments and relate the growth of dust conductivity to its amount using the specified non-contact capacitive dust amount sensor [53].

As noted, if the conductivity growth of carbon materials is facilitated by a ratio growth of the $sp^2$ states, then for tungsten oxides this is controlled by a decrease in the O/W ratio,

i.e., the creation of oxygen vacancies, which can be also controlled by XPS, IR, and Raman spectroscopies.

In practice, the W-dust composition detection efficiency using XPS in the WEST tokamak is reported in [53]. The Duster Box device for collecting dust using a controlled airflow is described, which was tested in Tore Supra (carbon-walled) and ASDEX Upgrade (tungsten-walled) tokamaks. The advantage of this collection method is that to explore different locations of plasma-facing surfaces, including dust collection on a divertor, dust samples were examined ex situ by XPS using the $W4f_{7/2}$ and $W4f_{5/2}$ doublet. The XPS spectrum revealed metallic W with binding energies of 31.0 and 33.2 eV, oxides $WO_x$ ($x < 2$) with peaks at approximately 31.7 and 33.9 eV, a small contribution of $WO_2$ at 32.4 and 35.5 eV, as well as a maijor contribution of the $WO_3$ phase at $W4f_{7/2} = 35.6$ eV and $W4f_{5/2} = 37.8$ eV (we think, with some O-vacancies, since the dielectric $WO_3$ phase indicates $W4f_{7/2} = 35.8$ эB, as shown above).

Another feature of W-dust, similar to the effects observed at the Princeton Large Torus (Princeton Plasma Physics Laboratory) and TEXTOR tokamaks, was found at the T-10 tokamak when working with W-limiters and a lithiated chamber. Tungsten dust during ohmic discharges led to the periodic accumulation of tungsten ions in the center of the plasma column within the framework of a repetitive cyclic process, which caused bursts of radiation losses on tungsten in the center of the plasma, and the periodic repetition of "small" disruptions in the discharge, as a rule, ended by the disruption of the discharge current [54].

As a result, the presented data on tungsten and carbon erosion products also demonstrate the effectiveness of IR, Raman, and XPS spectroscopic methods.

## 9. Unique Research on the Thermoplasmic Synthesis of Tungsten Dust

Important results were obtained using spectroscopic and other research methods, including nuclear reactions, X-ray diffraction, and TGA, showing "deeper" penetration in the bulk of the samples during the thermoplasmic synthesis of tungsten dust at the high heat flux system plasma facility (Institute of Plasma Research, Sonapur, India), demonstrating a high heat flux at the ITER level [27].

Thus, a target plate made of polycrystalline tungsten was irradiated with a laminar collimated jet of argon plasma with a flux of 9.3 $MW/m^2$ (350 A, ~10–20 kW, pressure in the chamber 20–400 mbar) for 30 min of irradiation, when the target temperature reached 4600 K, while the plasma temperature and density equaled 6900 K and $1.5 \times 10^{21}$ $m^{-3}$, respectively.

The dust studies using X-ray diffraction, XPS, BET, EDX, TGA, and electron microscopy, as well as the assessment of the hydrogen content using the resonant nuclear reaction $^{19}F(^{1}H, \alpha\gamma)^{16}O)$, etc., showed the following main results.

The melting of tungsten and its subsequent condensation lead to the growth of hierarchical vertical microstructures in the central open region, while polyhedral microparticles were deposited on the peripheral region. Both equilibrium polyhedral and nonequilibrium spherulite forms of tungsten crystals in the $\alpha$-phase were obtained (Figures 1 and 2 in [27]), —when the SEM electron microscopy image showed dumbbell-type spherulites and a dust morphology similar to those previously identified in tokamaks and in divertors simulating ITER. In addition, a non-equilibrium morphology, according to the authors, will also dominate in future powerful tokamaks.

At the same time, a cauliflower-like particle morphology was observed under all synthesis conditions at a low chamber pressure of 20 mbar, 350 A, and with a higher contribution of oxides and the β-W phase.

Measurements using the analysis of nuclear resonance reactions confirmed that tungsten dust particles in the nano- and micron range have a significantly higher hydrogen content compared to that of the same material in bulk form.

The authors noted that, under slow thermodynamic action, crystals grow into a macroscopic shape bounded by polyhedral surfaces with the lowest surface energy, which

under a limited microscope resolution may look like a spherical one. However, along with regular equilibrium particles, the authors also observed the formation of non-equilibrium crystal morphologies, which remained mixed with the former.

The higher specific surface area of the spherulitic microparticles results in enhanced gas uptake (including H and O), since the mesoporous crystals formed under high-power high-pressure synthesis conditions are responsible for the unusually high gas retention in dust samples. It grows up to 15 at.% for H (measured by the method of resonant nuclear reactions), and to 40 at.% for O (by recording gamma radiation in a nuclear reaction induced by protons, $^{18}O(p, p' \gamma)^{18}O$) at a specific dust surface area made by BET (Brunauer–Emmett–Teller analysis) = 38.1 $m^2/g$. The latter value becomes comparable to their content in carbon deposits during the erosion of carbon walls. As a result, the entire spectrum of samples of small tungsten particles demonstrates the absorption of hydrogen in large quantities, which are orders of magnitude higher than that for the bulk.

For comparison, it should be noted that the specific surface area of $CD_x$ flakes is close to ~30 $m^2/g$ (300 K) and ~200 $m^2/g$ (623 K) [17] at a content of (D + H)/C ~ 50–100 at.%.

The authors also carried out the oxidation of dust by heating in an oxygen atmosphere up to 800 °C, and the chemical composition of the samples' surface was verified by XPS, with a sensitivity of ~1 nm of the film thickness (photoelectron escape depth for W4*f*). TGA showed the onset of oxidation of W particles at approximately 200 °C, which at a higher temperature starts sublimating beyond 650 °C. It was noted that the surface oxidation of dust is an important issue, since the resulting oxide layer slows down the diffusion of hydrogen isotopes into subsurface regions, since H atoms can be trapped or dissolved in the formed oxides and carbides, such as $WO_3$, $W_2C$, and WC [55].

However, both an increase in the thickness of saturated layers during long-term tungsten irradiation with atomic hydrogen and a deep penetration of H into the bulk of the material at a diffusion coefficient of $\sim 10^{-6}$ $cm^2/s$ at 300 K, can lead to a significantly greater retention of H in W, and thus this reduces the advantage W as a wall facing the plasma, the authors indicate.

Atomic hydrogen H can also be captured by impurities such as C, O, and S, and the C and O atoms in W can react with H in the subsurface, creating complex chemisorbed impurities of these elements. Impurities tend to be embedded at grain boundaries such as WC, $W_2C$, and $WO_3$, and weaken the bond between adjacent grains of tungsten [55].

Since H can also be captured or dissolved in $W_2C$ and $WO_3$ with the formation of intercalated compounds with H atoms, this will further weaken the bond between W grains. In addition, there is the danger of deposition on the W surface of an impurity of carbon or hydrocarbons from the residual vacuum, which acts as a barrier against the emission of an H atom implanted in tungsten and easily desorbed from W, which will further increase the hydrogen retention in tungsten.

Similar safety considerations for W led to the development of self-passivating tungsten alloys by adding various oxide-forming elements that cause a formation of a protective oxide scale when exposed to air. For example, a significant improvement in oxidation has been achieved with the W–Cr–Y and W–Cr–Ti systems due to cold-spraying technology, when their oxidation behavior was improved by the addition of Cr content [56]. All coatings exhibited high hardness levels, good interface quality, and a potential for formation of stable $Cr_2WO_x$ phases.

Tungsten dust is known to oxidize even inside tokamaks because the vacuum is never perfect and the dust usually stays inside for a long time (except for small dust particles). XPS measurements showed the presence of W4$f_{7/2}$ metallic tungsten doublet peaks at a binding energy of 31.0 eV and W4$f_{5/2}$ at 33.2 eV, together with the $WO_3$ oxide peaks predominating in intensity, with W4$f_{7/2}$ at 35.8 eV and W4$f_{5/2}$ at 37.9 eV.

As seen from the above, the XPS technique for the in situ control of the erosion products in the form of films and dust is an important issue in maintaining the safe operation of fusion devices. Important to note is the depth of the photoelectron escape from metals

for XPS is ~1 nm, which makes it possible to effectively control not only the onset of dust oxidation, but also the onset of film growth.

Further, in the presence of an in situ control system of the type indicated by the MAPP, it will be possible to constantly monitor the film growth process using the XPS methods indicated in this system and the thermal desorption of accumulated gases using a quadrupole mass spectrometer, at any thickness of the erosion deposits, despite the small escape depth of the W4$f$ photoelectrons but taking into account the high efficiency of the XPS method and vibrational IR and Raman spectroscopy.

More precisely, this control for the case of the W-wall can be supplemented by IR spectroscopy of the vibrational modes W–H, W–O, etc., as was indicated, and both of these methods were also used in the case of CD$_x$ films from the T-10 tokamak.

Finally, to assess the huge jump in the level of knowledge between the period 1980–1990 and the present time, we note the following. According to experimental and theoretical studies of that period, hydrogen chemisorption on a clean W(100) face shows a complex series of changes in the W–H–W bridging bond that occurs with increasing coverage to the maximum, i.e., one or two monolayers, and most of the chemisorbed H is desorbed at temperatures above 500 K [55]. The injection of an energy plasma flow of H atoms greatly changes the properties of the surface and subsurface regions of W as a result of the defects' formation—vacancies, bubbles and interstitial loops, and as a result, the surface is deformed by swelling and the formation of bubbles, grooves, and new structures of the type considered above.

## 10. Progress in Tungsten Studies by Raman Spectroscopy

The IR vibrational modes of WO$_3$ films, the most common product of W erosion, are related to its structure. As reported, tungsten with a stable modification ($\alpha$-W) forms bcc crystals of the body-centered cube. An ideal perovskite (ABO$_3$ type) of the WO$_3$ unit cell is a cubic oxide, where W ions occupy the corners of the cell, O ions bisect the edges of the unit cell, and the central ion (A) is absent but can be filled with H, Li or Na ions after their intercalation, forming bronze (Figure 1 in [57]). In addition, each W ion is surrounded by six equidistant oxygen ions. These WO$_6$ octahedra, which form infinite chains, are articulated at common angles with O atoms at the vertices; point defects are eliminated, consisting mainly of oxygen vacancies.

WO$_3$ films deposited on substrates may be amorphous or polycrystalline. WO$_3$ films are amorphous at T < 250 °C, and become polycrystalline when heated above 400 °C. The amorphous $\alpha$–WO$_3$ film has a certain ionic and electronic conductivity, large open pores with square or hexagonal channels, and consists of clusters. The latter contain at least 3–8 WO$_6$ octahedra [57], which are linked by common corners or edges with W–O–W bonds [27] or hydrogen bonds through water bridges, with terminal W=O bonds on the cluster surface. The instability of the ABO$_3$ perovskite structure in the absence of the central A ion leads to the tilting of the WO$_6$ octahedra and the antiferroelectric displacement of tungsten atoms from the center of the octahedron, due to the strong electrostatic interaction between the W$^{6+}$ ions. Similar crystallographic distortions are responsible for the above mentioned five phase transitions in the temperature range from ~200 to 1700 K.

The most stable WO$_3$ phase at room temperature has a monoclinic structure but transforms into an orthorhombic or tetragonal phase at higher temperatures. The main differences between the phases are shifts in the positions of W atoms inside the octahedron and changes in the lengths of W–O bonds [27]. The observed vibrational bands mainly include vibrations of W=O, W–O, and W–O–W atoms, such as the W–O ($\nu$) stretching, W–O ($\delta$) bending, and O–W–O ($\gamma$) deformation modes, including those involving water vapor with O–H($\nu$, $\delta$) stretching and bending modes. The main vibrations of tungsten oxide occupy the region of 1453–400 cm$^{-1}$.

According to [58], there are several strong and weak bands at 417, 500, 670, 853 and 982 cm$^{-1}$. The 500 cm$^{-1}$ band is related to the strong bonding of the oxide lattice in the hydrated WO$_3$·nH$_2$O material. The band at 670 cm$^{-1}$ refers to the out-of-plane deformation

mode (W–O–W)$\gamma$. The shoulder at approximately 982 cm$^{-1}$ refers to the W=O end modes on the surface of the clusters, and at 853 cm$^{-1}$ there is the W–O–W bridge mode, the intensity of which noticeably grows after annealing. The poorly resolved wide band in the region of 3200–3600 cm$^{-1}$ and two peaks at 1453 and 1613 cm$^{-1}$ are due to the fact of moisture, and refer, respectively, to the $\nu$(OH) and $\delta$(OH) modes, which are detected due to the high sensitivity of W to the presence of OH -groups in IR spectra [59].

Stretching O–H modes are observed in broad bands at 3200–3600 cm$^{-1}$, which is close to these modes in CD$_x$ films. As a result, the main valence IR modes in CD$_x$ films, associated with the adsorption of hydrogen isotopes, are located at frequencies of 2300–2100 cm$^{-1}$ ($\nu$CD$_{2,3}$) and 3000–2800 cm$^{-1}$ $\nu$(CH$_{1,2,3}$), and they do not intersect with fundamental WO$_3$ modes in the frequency range 1453–400 cm$^{-1}$.

Further, in an innovative work [60], a large and important analysis of porous oxide and nitride films and nanoparticles in contact with tungsten plasma was performed using Raman spectroscopy with laser excitation at 514.5 nm, which showed exceptional efficiency. The effect of the surface roughness on the measured spectra was also revealed. There was used a wide range of laboratory samples grown by the pulsed laser deposition (PLD) with a glow discharge in argon. These are W films with different morphologies (porous or compact) and with different contents of O and N, W nanoparticles, W bulk sample, and thick WO$_3$ layers. Comparative data were also obtained for W, WN, compact W–O oxides, and porous W–O samples using XPS, as well as atomic force microscopy, scanning (SEM), transmission (TEM) electron microscopy, and DRS.

Due to the presence of hydrogen in tungsten oxides, the Raman spectrum is "rich" for oxides and bronzes, since there are optical phonons, as WO$_3$ has a strong electron-phonon coupling characteristic of perovskites, as mentioned. All this makes the Raman method more efficient for tungsten deposits with their rich Raman spectrum than that for hydrocarbon deposits.

Regardless of the stoichiometry, the O/W spectra mainly consist of bands associated with the bending and stretching W–O–W modes, in addition to the band associated with the surface. The main bands are 270, 714, and 805 cm$^{-1}$ of the three main Raman active modes for bulk monoclinic WO$_3$. For d-WO$_3$, $\alpha$-WO$_3$, and WO$_x$, a band at approximately 950–960 cm$^{-1}$ is observed, and the W–N bonds are at 471 cm$^{-1}$ and in the range of 700–800 cm$^{-1}$. Differences were also found in the spectra of WON samples, compact and porous. Oxides and nitrides can be detected and distinguished by the presence of additional broad bands in the range of 300–600 cm$^{-1}$.

The physically adsorbed H atom easily diffuses and integrates into the structure of WO$_6$ octahedra, which make up the erosion product WO$_3$ trioxide. The voids inside the WO$_3$ film with chains of octahedrons are the result of the random packing of clusters and mainly result in an open structure, which is usually filled with water molecules from the air or adsorbed gases, such as H, O, and N, which can easily diffuse inside these structures [48].

On the whole, the H atom can be captured in a chemical bond with C, O, and S impurities. In this case, the formation of OH groups is observed in the range from 1453 to 3600 cm$^{-1}$, and the intense peak at 1142 cm$^{-1}$ refers to the bending mode $\delta$(OH) in the W–OH group [48].

As a result, among the important results of the Raman spectroscopy of tungsten erosion products, the authors noted the following issues, which are of great practical importance for fusion devices:

- Raman spectroscopy is sensitive enough to detect surface natural W-oxides, and the optimal experimental conditions for such measurements have been found;
- For porous materials and nanoparticles, a larger proportion of tungsten oxides was found than for compact ones due to the high roughness of the first materials;
- The intensity of the Raman band changes and increases qualitatively with an increase in the O content;
- The crystallographic structure affects the intensity of the bands;
- Porous materials are less thermally stable than compact materials;

- This Raman spectroscopy method has the advantage of rapidly obtaining a large amount of data based on semi-qualitative spectroscopic observation and quickly characterize the morphology and chemistry of the analyzed samples. This can be used to select samples that need to be quantitatively analyzed later using more laborious analysis methods, such as XPS, AFM, SEM, TEM, EDX, TOF, and ERDA.

Diagnostics of in situ monitoring of films and dust in tokamaks during plasma discharges using the described methods of Raman, IR, and XPS spectroscopy is important from the point of view of the safe operation of fusion devices, especially at high thermal loads on the divertor. Indeed, the formation of liquid tungsten under hard plasma loads, is a serious problem, since critical plasma contamination with materials with high Z such as tungsten $^{74}$W, formed as a result of blistering and other processes, at a relative density W in the plasma above $\sim 10^{-5}$, can cause contamination of the central plasma and adsorption of tritium, leading to plasma disruption [61].

In addition, W-dust is hazardous due to the fact of its radioactivity during neutron irradiation with the formation of radioactive isotopes $^{185}$W and $^{187}$W, as well as toxicity and the ability to chemically react, and form a suspension and explode during emergency air intake as a result of loss of coolant. The formation of volatile $WO_3$ impurities at high temperatures is also hazardous, since they become radioactive when irradiated with neutrons [62].

From this point of view, carbon, according to the authors, is a much more "sparing" material than tungsten or other metals. At the early stages of ITER design, it was proposed as a protection against a very high heat flux onto a divertor with carbon fibers and reinforcement with pitch fiber, which has high thermal conductivity and high thermal stability, that is with almost twice the thermal conductivity than that for high-density tungsten grades. At present, this refers to the noted planned initial phase of the JT-60SA tokamak operation with carbon materials.

The similar problem of vacuum chamber wall erosion was found during the study of $CD_x$ films compared with the considered erosion of tungsten-containing materials. Thus, the control over the erosion of the stainless-steel wall of the T-10 tokamak chamber was performed using the XPS spectra of $CD_x$ films for the impurity doublet Fe2$p$ (711.4 and 724.8 eV) and Fe3$p$ (55.6 eV), corresponding to iron oxide $Fe_2O_3$ [63]; moreover, the survey spectrum also contained peaks of Cu2$p$ and Fe3$s$.

The appearance of an iron impurity inside the volume of thick $CD_x$ flakes ($\sim 20$–$30$ μm) in the amount of $\sim 0.7$ at.% was found using "deeper" or bulk research methods, such as X-ray fluorescence analysis with X-ray excitation at 21 keV [1,49], X-ray absorption of the EXAFS FeK edge (excitation energy up to 30 keV), X-ray diffraction ($\sim 10$ keV), and EPR spectra (9.9 GHz, or a wavelength of 3 cm)—demonstrating a broad line with $g \approx 2.07$.

As evident, the innovative studies in [60] on tungsten-containing erosion products, including spectroscopic Raman and XPS methods, are not only of great scientific but also of great practical importance.

## 11. Fractality of Erosion Products

### 11.1. Surface Fractality of Carbon Globular and Tungsten Films and the Hazadous Factors of W Erosion

In the T-10 tokamak, the film formation of various types was noted depending on the place of deposition, surface temperature, and plasma discharge mode [1]. Away from the graphite limiter, under discharge modes without overheating of the limiter, a smooth $CD_x$ film (flake) formed on the chamber wall at a temperature of 300–400 K. The film deposited near the limiter in the mode of the limiter heating up to $\sim 1000$ °C and at high fluxes of plasma particles is porous and looks like a cauliflower [1,64]. As a result, thick globular porous films [64] with a fractal (self-similar, or "self-repeating") structure of the "cauliflower" type were deposited near the graphite limiter and annular diaphragm of the T-10 tokamak under high plasma flux and under high temperature. This was also

established in experiments on the tokamaks DIII-D, TEXTOR-4, JT-60U, JET, etc. [1], and the size of the globules was in a wide range from ~10 nm to ≈20 μm [65].

It is important that the thickness of the globular films was up to 150 μm, the accumulated deuterium value was D/C = $4 \times 10^{-4}$ (which is three orders of magnitude lower than that of $CD_x$ films, for D/C ~ 0.5), with the presence of Fe, Cr, and Ni metal impurities, and the inhomogeneous distribution of deuterium [66].

For the majority of globular films studied using a scanning tunneling microscope with a resolution of ~10 nm, when analyzing the surface of the films on a scale from ~10 nm to ~10 μm and a scanning electron microscope, there is a power dependence of the size distribution of the number of globules $N \sim r^{-D}$, which is typical for a fractal structure [1], where the value of the fractal dimension of the surface equals $D = 2.2 \pm 0.2$. More precisely, this D value is characteristic of the transition from a structure with the almost smooth boundaries of a flat disk ($D = 2$) to a structure consisting of aggregates with a highly developed fractal (or rough) surface. The size distribution of dust particles in the T-10 tokamak and the LHD stellarator of carbon dust [64], which is formed during the operation of a graphite limiter in plasma discharges, has a similar fractality of $2.2 \pm 0.1$.

The considered amorphous, or so-called "disordered" structures in fusion devices show self-similarity, not trivial disorder, when the surface structure hierarchy is described by a power law [64].

The fractal analysis of surface roughness (nanoprotrusions) was not performed with $CD_x$/Si(100) thin films from T-10, as was conducted, for example, in [67] with a-C:H:Fe/Si(100) thin films deposited by chemical vapor deposition using RF plasma. Studies of 3D images of the surface of these samples with an atomic impurity fraction of Fe/C = 0.007–0.06 and a high proportion of $sp^3 \approx$ 50–70 at.% using an atomic force microscope, showed the surface fractal dimension of nanoprotrusions $D$ = 2.26–2.32, which is typical for a poorly developed surface with large smooth grains on it, and this is quite close to the indicated surface fractality of globular films of the "cauliflower" type. It should be noted that in order to work with a tunneling microscope, these globular films must have conductivity (the study of which was not reported in [64]), in contrast to the described dielectric properties of $CD_x$ flakes [49].

This also indicates the difference in their electronic structures due to the differences in the plasma action and temperature on the eroded carbon materials. It is known that as a temperature of the deposited film increases, the fraction of the $sp^2$ states increases, i.e., the conductivity increases and the value of the band gap $E_g$ decreases for conductive graphite ($sp^2 \approx$ 100 at.%) at the value of $E_g$ ~ 0.

Thus, globular films deposited at high temperature should have a high fraction of $sp^2$ states, which is also indicated by a small fraction of D/C ~$10^{-4}$ for adsorbed D atoms on dangling bonds of $sp^3$ C–. Therefore, measurements can be performed using a tunneling microscope, and in a wide range of three to four orders of magnitude of the structure sizes.

A similar fractal structure of the "cauliflower" type was also observed on tungsten materials. For example, when materials such as tungsten, beryllium, and carbon were irradiated by high-temperature plasma pulses at a QSPA-T plasma accelerator, a fractal surface structure was also observed with a fractality of $2.2 \pm 0.1$ on scales from nanometers to tens of microns [64]. We assume, that for the W-erosion product of the $WO_3$ type, this means the presence of a large ratio of vacancies (i.e., $WO_{3-x}$), as well as the possible participation of another conductive erosion product, $WO_2$.

This points [64] to the universal mechanisms of the stochastic clustering of materials under the influence of high-temperature plasma with strong turbulence of the near-wall plasma, which leads to the instability of the flows of deposited ions and molecules on the surface. At the same time, experimental observations in the LHD stellarator showed that amorphous globular films are not formed in a glow discharge. The formation of a fractal structure of the "cauliflower" type on a number of materials, including carbon and tungsten, which have a large specific surface, and are capable of adsorbing a large number of hydrogen isotopes (including tritium) in fusion devices, as well as the formation, as

indicated, of tungsten "fuzz" under high flows of helium plasma, pose a serious problem to radiation safety in the ITER program [64].

Thus, when a W-plate was irradiated at the PLM facility (MEPhI, Moscow, Russia) in a helium plasma flux of $3 \times 10^{18}$ m$^{-3}$ at a thermal load of up to 1 MW/m$^2$ and a temperature above 1000 °C, surfaces of "fuzz"-type nanostructures formed on the tungsten, with W fibers up to 50 nm in diameter and a nanostructured fuzz layer depth of up to 1.6 μm [68].

The hazard of their appearance lies in the fact that arc discharges develop better on such a surface, and the erosion product in the form of "fuzz hairs" is the most dangerous dust, which will lead to quenching of the DT reaction in the fusion reactor as a result of the plasma energy loss due to the bremsstrahlung and recombination on these dust particles. The situation is also aggravated by the fact that the energy threshold of He ions for knocking out a W atom from the surface layer with the formation of clusters—, or "fuzz hairs", is ~30 eV, which is three times lower than the W sputtering threshold.

The bulk electron structure of globular carbon films of the "cauliflower" type was not studied, for example, using X-ray scattering methods (with an absorption depth of up to 10,000 μm in carbon and ~10 μm in tungsten); only their surface structure was studied using scanning tunneling and electron microscopes.

*11.2. Bulk Fractality of Carbon Smooth CD$_x$ Films*

Studies of the bulk structure of smooth amorphous hydrocarbon flakes CD$_x$ ($x$ ~ 0.5) with a thickness of ~20–30 μm, deposited from working plasma discharges of the T-10 tokamak deuterium plasma and cleaning discharges (i.e., under relatively "soft" plasma exposure) in contrast to globular films of the "cauliflower" type were carried out using the methods of small-angle and wide-angle elastic X-ray scattering [2]. The SAXS + WAXS measurements of goldish flakes were performed in the Debye–Scherrer transmission geometry at the Structural Materials Science beamline of the Kurchatov Synchrotron Radiation Center (Moscow) using FujiFilm Imaging Plates as a detector. Additionally, φ neutron diffraction pattern was measured using a multi-detector constant-wavelength neutron diffractometer DISK at the IR-8 neutron source (NRC Kurchatov Institute, Moscow).

SAXS + WAXS studies refer to elastic x-ray scattering on the electron density, when the electronic structure of the scattering objects is studied, which is especially important for nonperiodic disordered systems. In this case, synchrotron radiation is used in a wide range of scattering vectors $q$ = 0.04–100 nm$^{-1}$, where $q = 4\pi\sin\theta/\lambda$, $2\theta$ is the scattering angle, $\lambda$ is the wavelength of the X-ray radiation with an energy of 9–26 keV, and the maximum depth of the X-ray absorption reaches ~$10^4$ μm for carbon.

In this case, the $q$ value is related to the size of the studied scattering object $d$ approximately as $q$ ~ $1/d$ and covers, as measurements have shown, the region of structure elements with sizes from ~1 to 60 nm, where the limit of the registration of the structure size from above and below is determined by the conditions of the experiment using the SAXS + WAXS method, i.e., due to the fact of X-ray ultra-small incidence angles and extra-large scattering angles, respectively.

The experimental curve of small-angle X-ray scattering in the range of $q$ = 0.04–2.6 nm$^{-1}$ (Figure 5) is described by a power law, when the intensity $I(q) \sim q^{-a}$, which indicates fractality, where $a$ is the size of the fractality, as in the case of the considered carbon and tungsten objects with a "cauliflower"– type surface structure with a fractality of 2.2 ± 0.1. In this case, as $q$ grows from the maximum resolvable scale $L$ to tens of nanometers (at $L$ ~ $1/q$), ever smaller and finer structural elements are resolved, and the shape of the observed scattering curve can be considered as a result of adding partial scattering curves from particles of different sizes in a polydisperse system.

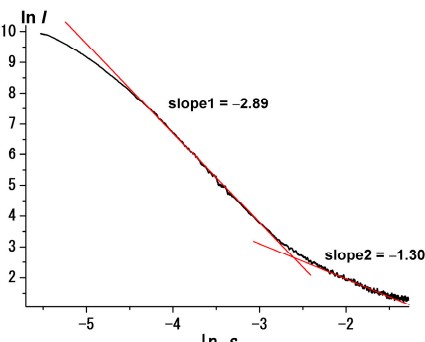

**Figure 5.** The SAXS spectrum of a gold $CD_x$ flake at the scattering vector range $q$ = 0.04–2.6 nm$^{-1}$ in a double log scale. The Guinier region extends from $q_0$ up to ln$I \sim R_g^2 q^2$.

The SAXS scattering curve $I(q) \sim q^{-a}$ can be approximated in a double log scale by two slopes $\Delta(\ln I)/\Delta(\ln q)$, showing two slopes in the range of $q \approx$ 0.1–2.6 nm$^{-1}$: $a$ = 2.89 ± 0.1 and $a$ = 1.30 ± 0.1. The absence of singularities and the monotonic decrease in the scattering intensity, which is almost linear in double logarithmic coordinates in two areas, means that the samples under study can be considered fractals (self-affine structures).

In the SAXS region, extending over $q$ changes less than one order of magnitude, i.e., the limits of fractality in the space of dimensions $L \approx 1/q$ extend for less than one decade. However, such a limited fractal size range from 0.5 to 2 decades is quite typical for many amorphous structures, including carbon ones [2].

Another approach can also be applied to this SAXS curve, in view of the fact that the fractal region is rather short both in terms of the intensity scale and in terms of the scattering vector scale. In such a case, the so-called unified scattering function model [69] is often applied. This means modeling of a part of the experimental SAXS curve $I(q) \sim q^{-a}$ in the limited region of $q$ = 0.04–0.5 nm$^{-1}$ (Figure 6) using the universal scattering function, having the form of a sum of the Guinier region (extends from $q_0$ up to ln$I \sim R_g^2 q^2$), and the power-law region at higher $q$ values: $I(q) = G \exp(-q^2 r_g^2/3) + B\{[\text{erf}(q r_g/6^{1/2})]^3/q\}^p$, i.e., it is a unified function with one level of structure, where $g$, $b$, $r_g$, and $p$ are fitting parameters.

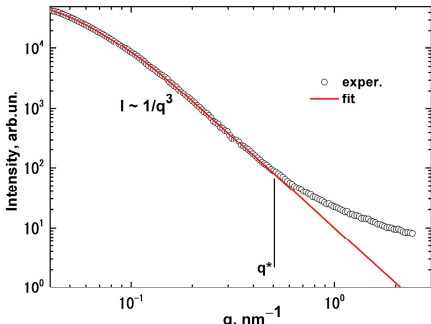

**Figure 6.** The experimental SAXS curve modeled by the unified scattering function in a shorter range of $q$ = 0.04–0.5 nm$^{-1}$ [70]. The model starts to deviate from the experimental curve at q = q*.

Fitting this short range leads to the fractality size $a$ = 3.04 ± 0.05, which shows a structure close to the previous approach of the intermediate case between the formation of bulk (2 < $a$ < 3) and surface (3 < $a$ < 4) fractals with an uneven, rough surface, which is typical in the physics of plasma polymers which include soft a-C:H films. Such a slope is characteristic of an intermediate case between mass-fractal and surface-fractal behavior [71]. In this case, one can obtain the largest recorded size of fractal aggregates, characterized by the Guinier radius of inertia (radius of gyration $r_g$), equal to $r_g$ = 32.5 ± 0.3 nm, i.e., the total maximum fractal size equals ~60 nm. This value is caused by a limited instrumental resolution for ultra-small SAXS scattering angles. A minimum size $q$* can be estimated as

inverse momentum at which the model starts to deviate from the experimental curve, as seen in Figure 6. This shows a size $1/q^* = 2$ nm, which corresponds to the mean distance between primary particles building up the fractal aggregates.

Thus, the first SAXS fractality value $a = 2.89$ characterizes the nonperiodic amorphous structure of a mass or volume fractal with a rough boundary, which is typical for systems with a branched carbon network in 3D space, including branched polymers. Moreover, other known fractalities are as follows: $a = 4$—smooth boundary 3D scatterer, $a = 3$—surface fractals with a rough surface, $a = 2$—thin disk with smooth borders, $a = 1$—long thin rod. The second SAXS fractality $a = 1.30$ can be considered a gradual transition from the scattering region of a mass fractal ($2 < a < 3$) to the scattering region of a long thin rod with $a = 1$, which is typical for linear structural elements forming a branched 3D carbon network.

Thus, in the branched carbon network of $CD_x$ films, there are such structural elements as linear $sp^3$ C–C, C–H(D), C–O, O–H chains showing a fractality $a = 1$, $sp^3$ C– free radicals, $sp^2$ olefin chains C=C, planar benzene rings $sp^2$ C=C (fractality $a = 2$) and others, according to the experiments conducted using IR and XPS spectroscopy.

In the experimental WAXS region at $q = 3$–100 $nm^{-1}$ in Figure 7 (as a continuation of the SAXS curve in Figure 5), model calculations were used to best reproduce the shape of the experimental WAXS curve using the Debye formula. The experimental WAXS curve reveal three major peaks, fitted by Gaussians, centered at $q = 9.55$, 25.9, and 53.1 $nm^{-1}$. Their large widths clearly imply a noncrystalline atomic arrangement within the carbon flakes. Figure 7 also shows the neutron diffraction curve for the same flake for comparison. Quite surprisingly, the neutron scattering curve represents peaks only at 25.9 and 53.1 $nm^{-1}$.

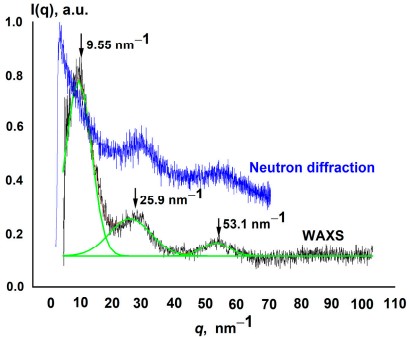

**Figure 7.** Experimental WAXS curve at $q = 3$–100 $nm^{-1}$ (with 3 Gaussians peak fittings) and a neutron diffraction curve of the same $CD_x$ flake.

We believe that the reason for the absence of the first peak at 9.55 $nm^{-1}$ in the neutron diffraction curve is due to the low contrast in the neutron scattering on the C–D bonds against the "background" scattering on the C–C bonds of the carbon network. The low C–D contrast is due to the closeness of the coherent neutron scattering cross-sections for atomic deuterium and carbon, which are, respectively, $\sigma(D) = 5.59$ barn and $\sigma(C) = 5.51$ barn (1 barn = $10^{-24}$ $cm^2$).

At the same time, molecular deuterium was not detected in the $CD_x$ films in the Raman spectra, as well as molecular hydrogen, in the modes 4160 ($H_2$), 2993 ($D_2$), and 3630 $cm^{-1}$ (HD). Since D was found in the C–$D_{2,3}$ vibrational modes (see Figure 3) only in the $sp^3$ state, the peak at 9.55 $nm^{-1}$ refers to the $sp^3$ C–$D_{2,3}$ states or to atomic deuterium in the interstitial space.

For tritium $\sigma(T) = 2.89$ barn, its presence in C–T scattering structures can be detected in neutron and X-ray diffraction patterns. Hence, it can be assumed that the first peak at 9.55 $nm^{-1}$ is due to the presence of deuterium D, and the second and third peaks are associated with atomic hydrogen.

These peaks observed both in X-ray and neutron scattering curves are rather typical of carbon materials. Formally they can be assigned to (10) and (11) two-dimensional reflections of a graphene layer and thus they are inherent to a hexagonal arrangement of $sp^2$-hybridized

carbon atoms close to that in graphene. Meanwhile, the large widths of the peaks indicate very tiny ordered domains of only a few carbon hexagons. Therefore, an attempt was made to reproduce the shape of these singularities using the Debye formula [70]. The latter describes a discrete sum of elementary scatterers for which the coordinates of atoms within the selected cluster are known, and the diffraction scattering intensity $I(s)$ for the scattering vector $s = 2\pi\sin\theta/\lambda$ is taken from the sum of the scattering amplitudes $f_i$ from the i-th atom and $f_j$ from j-th atom, where the summation is taken over all interatomic distances $r_{ij}$ between atoms in positions $r_i$ and $r_j$, i.e.,

$$I(s) = \sum f_i(s)f_j(s)\sin(sr_{ij})/(sr_{ij}).$$

As a result, the model calculations using the WAXS curve with three broad peaks resulted in a structure of the so-called minimal fractal aggregate $9 \times C_{13}$ formed by nine fragments in 3D space, each consisting of three interconnected $sp^2$ C=C benzene rings with a size of $0.75 \pm 0.15$ nm, forming an ensemble $C_{13}$ consisting of $sp^2$-clusters, or with thirteen carbon atoms forming three fused benzene rings (Figure 8). It appeared that the model "$9 \times C_{13}$" of atoms is sufficient to reproduce the widths and relative intensities of the three experimental scattering peaks on the bottom curve in Figure 8.

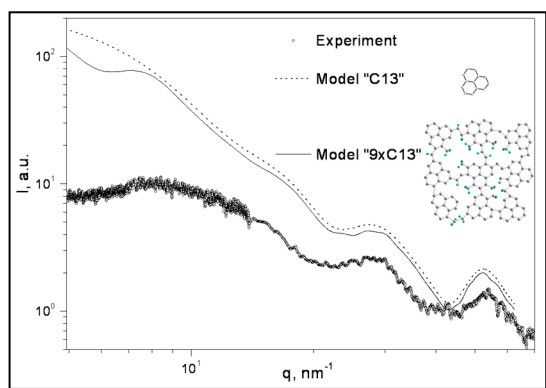

**Figure 8.** Simulation of WAXS patterns with 3 wide peaks for two simple patterns "C13" and "$9 \times C13$" with corresponding curves calculated using the Debye equation, and displaying the minimum fractal aggregate "$9 \times C13$". The bottom curve is experimental.

This can be confirmed by the X-ray diffraction patterns of $CD_x$ flakes taken at the STM station (National Research Center "Kurchatov Institute") at a wavelength of $\lambda = 0.1072$ nm in the Debye–Scherrer transmission geometry. The X-ray diffraction pattern consists of two main broad peaks, which correspond to interplanar spacings of 0.70 nm and 0.23, i.e., close in size to one and three benzene rings [63]. This is consistent with the smallest size of the initial fractal particles ~1 nm, corresponding to the above (Figure 6) model curve of the universal exponential-power scattering function obtained in the framework of the fractal model of polymers with a branched grid for modeling the experimental WAXS curve. In this case, the maximum recorded fractal polymer coil-type aggregate in 3D space has a size of ~60 nm, and connecting linear elements of the carbon network. These graphene-like $sp^2$-clusters are interconnected and form a defective lattice with vacancies in 3D space, since scattering with a fractality $a = 2.89$ in the region of scales $L$ of tens of nanometers (at $L \sim 1/q$) is not far from the fractality $a \approx 3.0$, which is characteristic of scattering from broken polymer chains in polymer physics.

The voids of strongly deformed graphene model layers (Figure 8) can be filled with D and H atoms and other structural elements that saturate broken C–C bonds. Further, in the specified defect lattice with vacancies, unsaturated chemical bonds are filled with adsorbed atoms D, H, linear $sp^2$ (C=C and C=O), and $sp^3$ elements—C–C, C–H(D), $C–D_2$, $–CH_3$, C–O, O–H, COOH, and $C_xH_y$ recorded in the IR spectra of $CD_x$ flakes and C1$s$ XPS spectra. The structure and size of the present fractal aggregates are confirmed by studies

on photoluminescence, electron paramagnetic resonance and thermal desorption mass spectroscopy [2].

The results obtained characterize the structural features of red and golden $CD_x$ flakes as a system with $sp^3 + sp^2$ states (ratio of $sp^3 > 60$ at.%), forming a branched 3D carbon network with a fractal (self-affine, or self-repeating) structure and high specific surface area [17]. This structure promotes the accumulation and storage of hydrogen and hydrocarbon isotopes in a stable chemisorbed state. However, the sorption properties of these films can be reduced due to the iron impurities present in the bulk (<1 at.%, from erosion of the SS chamber walls), the catalytic effect of which reduces the thermal desorption threshold of $D_2$ and $H_2$ [19].

In general, the structure of these $CD_x$ flakes represents a complex amorphous hydrocarbon–deuterium system with a high atomic content of hydrogen isotopes $(D + H)/C \approx 1$. As a result, the actually obtained multifractal cluster model of smooth $CD_x$ films with a high atomic content of hydrogen isotopes $x \approx 0.5$–$1.0$, confirmed by extensive experimental studies, reports the bulk 3D electronic structure and properties of these films from minimal fractal aggregate at the ~1 nm to ~60 nanometers level of linear dimensions (caused by limited instrumental resolution for SAXS ultra-small angles).

On the contrary, the surface 2D structure of carbon films of the "cauliflower" type with a fractality of $2.2 \pm 0.1$ and a low content of hydrogen chemisorbed isotopes ~$10^{-3}$–$10^{-4}$ was studied using scanning tunneling and electron microscopes in the wide range of linear sizes from ~10 nm to ~10 μm. Its structure was modeled using the mechanisms of stochastic clustering of materials under the influence of large fluxes of high-temperature plasma with strong turbulence in the near-wall region, as well as at high temperature of the T-10 limiter, which leads to the instability of the fluxes of deposited ions and molecules on the surface and to the formation of this structure [64].

Thus, the considered so-called "disordered" structures in fusion devices have not a trivial disorder, but self-similarity, when the hierarchy of the surface or bulk structure is described by a power law with different fractality values depending on the plasma exposure conditions.

## 12. Conclusions

(A)   Extension of the erosion products diagnostics methods from carbon to tungsten

It is shown that the technique for studying $CD_x$ hydrocarbon films developed in the past one–two decades and consisting of a dozen experimental techniques have shown that the properties of the films are, in fact, a "passport" of the processes of their formation under the action of the plasma conditions of the T-10 tokamak [9]. These methods have a universal character and, therefore, can also be effectively used for the analysis of erosion products of tungsten-containing materials.

In particular, this conclusion is true for the most common methods of Fourier-transform infrared spectroscopy, Raman scattering of light, X-ray photoelectron spectroscopy, thermal desorption spectroscopy, and X-ray absorption spectroscopy on a par with other methods used in the study of $CD_x$ films. As in the case of $CD_x$ films, examples of the application of a wide range of techniques for studying plasma–surface interactions, suitable for both carbon and metallic materials of the first wall, are presented. The considered new methods, such as ion scattering and analysis of various nuclear reactions involving light ions, significantly complement the arsenal of research to assess the accumulation of hydrogen isotopes H, D, and T in erosion products. In the case of studies on the oxygen accumulation in the erosion products of tungsten, W-oxides and WC, which are also hazardous, scanning electron microscopy, transmission electron microscopy, laser breakdown spectroscopy, energy-dispersive X-ray analysis, and AFM and STM probe methods are also useful.

(B)   New in situ control methods [28–34]

Of particular interest are the available, albeit few, developments of in situ control methods in the system of the MAPP (materials analysis particle probe) for analyzing first

wall erosion products during the operation of tokamaks, performed at a high level of plasma energy. Here are used XPS technique, ultraviolet photoelectron spectroscopy, photoelectron spectroscopy with angular resolution, ion scattering, direct recoil spectroscopy, thermal desorption spectroscopy, investigation of the conductivity of films and dust, and laser breakdown spectroscopy to determine the content of hydrogen isotopes H, D, and T in plasma-facing components of fusion devices in the JET and ITER.

These methods of in situ control make it possible to constantly monitor the dynamics of the growth of the deposited erosion layer in the intervals between plasma pulses despite the predominantly surface character of deposits. Unfortunately, we did not find in the literature information on the development of in situ methods such as Raman, IR, and energy-dispersive X-ray analysis, which could also be required for working with tungsten and other materials.

(C)   New ex situ diagnostic methods

In the absence of an in situ control system, it is possible to observe the process of deposit growth using the system of the input and output of samples with the studied irradiated materials at the level of the chamber wall for subsequent ex situ analyses, as, for example, was conducted with $CD_x$ thin films on Si(100), irradiated only by plasma discharges at the T-10 tokamak. At each new stage of the growth of the deposited layer, it can be reintroduced into the reactor, irradiated, and extracted outside for further ex situ studies using various methods, including XPS, IR, and Raman.

In addition, the methods for studying $CD_x$ films using X-ray excitation from ~10 to 30 keV SR source, nuclear reactions involving light ions with an energy of ~MeV, and EPR (including the case of $WO_3$ trioxide) mentioned for the JET erosion products can also be used when working with tungsten up to a thickness of ~10 microns, and neutron diffraction, TDS and TGA—up to tens of microns and more.

Furthermore, to study thicker (cm) layers, there is neutron activation analysis, which is used in metallurgy, geology, biology, etc. The identification of the desired elements occurs using nuclear $n\gamma$ reactions.

(D)   Similarity of the microstructuring of carbon and tungsten erosion products

It has been established that the considered so-called "disordered" carbon and tungsten structures of erosion products in fusion devices do not have a trivial disorder but possess self-similarity when the structure hierarchy (surface or volume) is described by a power law with different fractality values depending on the conditions of the plasma exposure. For the linear surface structures of carbon and tungsten deposits, a surface fractal structure extends from the subnanometer scale up to ~10 µm. However, for smooth $CD_x$ films, a new level has been reached due to the study of the bulk multifractal three-dimensional electronic structure using X-ray scattering SAXS and WAXS within linear dimensions from a minimum fractal aggregate of ~1 nm to a maximum cluster of ~60 nm. The latter dimension is characteristic of a fractal polymer coil-type aggregate, and its maximum value is due to the limited instrumental resolution for ultra-small SAXS angles.

(E)   Diagnostics needed for the safe operation of fusion facilities with metal plasma-facing components [60]

A comparative analysis of the spectroscopic studies of carbon and tungsten deposits makes it possible to identify the problems of the safe operation of thermonuclear fusion reactors. In our opinion and in accordance with the presented examples of work with a metal first wall, special attention should be paid to a number of hazards during the accumulation of erosion products and, first of all, the following main activities should be carried out when operating fusion devices:

E1. Continuous monitoring of the accumulation of hydrogen isotopes H, D, T, and O (in the case of tungsten) using IR, Raman, and TDS methods, as well as methods of nuclear reactions and laser breakdown spectroscopy.

E2. Constant monitoring of the accumulation of tungsten dust, including the formation of tungsten oxides, using IR, Raman, and XPS methods. Thus, spherulite W microparticles with a higher specific surface area formed under synthesis conditions under high-power and high-pressure plasma fluxes lead to enhanced gas absorption, and they are responsible for the unusually high gas content in W-dust samples, up to 15 at.% for H and 40 at.% for O [27], which is similar to the situation with carbon wall erosion leading to high hydrogen accumulation.

It is important to be aware of the hazard of W-dust due to the fact of its toxicity, chemical reactivity, ability to form dust suspension, and its explosion during an emergency air supply. In addition, the formation of volatile $WO_3$ impurities at high temperatures is hazardous, since they become radioactive when irradiated with neutrons. At the same time, Raman spectroscopy showed good sensitivity for the detection of surface W oxides, and the intensity of the Raman bands increases with an increasing O/W content.

E3. In the case of the presence of impurities of carbon deposits and dust, they can be identified in three ways: Raman spectrum (modes D and G, as well as by the growth of the background at the same time with the shift of its maximum towards higher wavenumbers in the case of a large accumulation of H and D), IR spectrum according to the indicated $sp^3$ $CD_{2,3}$ vibrational modes at approximately 2200 cm$^{-1}$ and others, and using XPS and XAES of the C1$s$ spectrum, with the possibility of estimating the $sp^2$ and $sp^3$ fractions and various impurities in carbon deposits.

E4. Periodic monitoring of the appearance of impurities other than those expected (using XPS and EDX).

Thus, the technique of the conducted comparative studies on tungsten-containing elements and $CD_x$ hydrocarbon films, including spectroscopic methods, was used and supplemented with new approaches for the effective study and control of the deposition of tungsten-containing materials of the tokamak first wall.

**Author Contributions:** Conceptualization, V.G.S. and B.N.K.; methodology, N.Y.S.; software, N.Y.S.; validation, V.G.S., N.Y.S. and B.N.K.; formal analysis, V.G.S.; investigation, V.G.S. and B.N.K.; resources, N.Y.S. and B.N.K.; data curation, V.G.S.; writing—original draft preparation, N.Y.S.; writing—review and editing, V.G.S. and B.N.K.; visualization, V.G.S.; supervision, B.N.K.; project administration, V.G.S. All authors have read and agreed to the published version of the manuscript.

**Funding:** This research received no external funding.

**Institutional Review Board Statement:** Not applicable.

**Informed Consent Statement:** Not applicable.

**Data Availability Statement:** The data presented in this study are available on request from the corresponding author.

**Acknowledgments:** The authors thank L. N. Khimchenko (National Research Center Kurchatov Institute, Moscow) and N. S. Klimov (TRINITI, Troitsk, Moscow) for providing samples from the T-10 tokamak and the QSPA-T accelerator, Y.V. Zubavichus for the help in obtaining the SAXS + WAXS spectra of $CD_x$ films, and also A.E. Gorodetsky (A.N. Frumkin Institute of Physical Chemistry and Electrochemistry Russian Academy of Sciences, Moscow) for valuable discussions and comments, and V.P. Budaev and A.B. Kukushkin (National Research Center Kurchatov Institute, Moscow) for the helpful comments.

**Conflicts of Interest:** The authors declare no conflict of interest.

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
