# Peer review of "Comparative Analysis of Spectroscopic Studies of Tungsten and Carbon Deposits on Plasma-Facing Components in Thermonuclear Fusion Reactors"

_symmetry, doi:10.3390/sym15030623_

Round 1

Reviewer 1 Report

Dear authors, 

your work is certainly very extensive.

You employed a large number of experimental samples to performe an interesting comparison between spectroscopic analyses of the component tiles of the walls exposed to the plasma of thermonuclear fusion reactors.

The experimental line is the comparison of spectroscopic methods for the identification of the erosion products of the tungsten coating components (films, surfaces, powder) with those of the carbon deposits, considering the carbon deposited in the vacuum chamber during the erosion of the graphite limiters in the T-10 tokamak or also formed by irradiating a tungsten target with an intense H-plasma in the QSPA-T plasma accelerator.

The authors have carried out a very laborious collection. Unfortunately this collection is assembled in a review complicated to read. The paper does not seem properly organized as a scientific article but as a term paper, starting from the choice of the summary or from the title that has been given to the paragraphs.

Small annotations must also be made to some parameters, abbreviated in the text and not specified: these may be unknown to the reader not expert in the each mentioned aspects (for example line 73).

Moreover, the introduction paragraph, which frames different operational scenarios, does not describe them in their mutual differences. The various sections seem unrelated to each other, both when the different facilities are proposed and when the different spectroscopic methods are exposed. Still other sections are not directly comprehensible (eg sentences on lines 172-178).

In the overview of the paragraphs, it is not clear why the "materials and methods" are made explicit only for a specific scenario, but above all, the figures and images of the configurations mentioned, or the diagrams of the positioning of the samples, are missing.

A description of the "operating methods" is then added in each of the paragraphs following the paragraph of the first results. The intent of the authors was certainly to sort out the different types of investigation, and for each investigation they compared the type of measurement that was conducted to obtain the analysed result. However, this approach requires a strong graphic and schematic impulse being an article 36 pages long.

Starting from the introduction paragraph, it is essential to insert a table that organizes the used measurement techniques and the cases they were used (which samples and why the choice).

It is not clear what the authors' direct contribution to the experimental activity, measurement and analysis, was (and therefore what the original contribution of the manuscript is, which is presented as a review). The reason for the choice of the operating methods relating to the aspects of the comparison (availability?) is also missing.

The list proposed in lines 243-268 is not suitable for the form of a scientific review.

Throughout the text of the manuscript, periods are repeated using impact sentences, such as the one on line 269 (there are also some further on, for example on lines 504, 512, etc.). It's not a report or a minute so I don't find the expressive way suitable.

A schematic is also needed in the next section from lines 275 to line 292.

In paragraph 3 two objectives of the work are finally exposed but it is about page 6, therefore too late. I suggest extensive restructuriation of the manuscript. This same paragraph, at point 3.1, in any case, is meager and without evidence. On the contrary, section 3.2 is particularly heavy and needs a change both in the name of the paragraph and in the organization of the text. If the authors want to insert a considerable amount of informations they have to restructure the manuscript by schematizing them, inserting tables and adding specific references on the need to use the operational methodologies used.

The entire paragraph 3 consists of 5 pages, up to page 11, and is not very fluid. Reading is particularly difficult because it is not aided by graphical representations.

In paragraph 4 recurring expressions are used which tend to underline aspects that do not require notification.

Perhaps much of the information that authors insert, assigning equal importance in the text that does not flow, can be moved to a section of supplementary content.

In paragraph 5 the first table appears (we are even on page 15), but this paragraph too is enriched with information that distracts the reader, and then references to small mentions are missing (for example line 628, line 638 and line 690). For details on the method I asked to discuss the observations made on the limits of the Raman technique applied to the specific case.

Again, paragraphs 6, 7, 8, 9, 10 and 11.1 make up another 10 pages without a priority scheme, a graphic figure or a data plot.

Suddenly paragraph 11.2 is enriched with a technical exposition different from the previous ones.

An excellent conclusion paragraph on the reported comparative studies must be credited to the authors.

Even the 71 bibliographic references are quite up-to-date and selected (apart from the previous reports).

I believe that a complete and significant revision of the manuscript is necessary in terms of lightening, driven towards the need to clarify the complexity of the operational and experimental references.

Reviewer 2 Report

please see the attached pdf

Author Response

Please the attached file

Reviewer 3 Report

The paper on Comparative analysis of spectroscopic studies of tungsten and carbon deposits on plasma facing components in thermonuclear fusion reactors give a review of studies of erosion products of the tungsten plasma facing components for thermonuclear fusion reactors.

The employed spectroscopic methods are similar with those used in analyses of carbon deposits and the authors compared obtained results. 

They gave detailed review for analyses of carbon-deuterium CDx (x ~ 0.5) smooth films, deposited at the vacuum chamber during the erosion of graphite limiters in the T-10 tokamak; mixed CHx-Me films formed by irradiating a tungsten target with an intense H-plasma flux in the QSPA-T plasma accelerator. 

They numbered different developed technique for studying the CDx films with several methods, including spectroscopic methods such as XPS, TDS, EPR, Raman, FT-IR. Also, they concluded that the methods are universal and can be supplemented by a number of new methods for tungsten materials, including in situ analysis of the MAPP type using XPS, SEM, TEM, probe methods and nuclear reaction methods. 

At the end the give detailed review on analyses of the fractality of the CDx films using SAXS+WAXS is compared with the analysis of the fractal structures formed on tungsten and carbon surfaces under the action of high-intensity plasma fluxes. 

In conclusions the authors recommended diagnostics needed for safe operation of fusion facilities with a metal plasma facing components. Also the concluded that a comparative analysis of the spectroscopic studies of carbon and tungsten deposits makes it possible to identify the problems of safe operation of thermonuclear fusion reactors.

In my opinion the paper represents a comprehensive review study on deposits on plasma facing components in thermonuclear fusion reactors. This type of study is  rare and very worth for thermonuclear fusion reactors facilities.

I have no specific objections. 

Reviewer 4 Report

The manuscript contains an extensive overview of the spectroscopic methods employed to study the effects of erosion by high-temperature plasma on metallic first walls of fusion devices, with reference to the expertise gained at the T-10 tokamak. Conclusions are potentially of interest for upcoming devices, JT-60 SA, ITER and DEMO. 

The argument is fairly specialized, will be of interest only to a restricted readership, but it appears dealt with thoroughly. I have no remarks about the content. 

The quality of the presentation is good but for some minor issues. There are not serious errors with the English, but some minor mistakes, likely due to sentences written hurriedly and not re-read carefully.

Since these mistakes do not prevent the manuscript to be read and understood, I do consider the paper acceptable in the present form, but recommend the authors to check the whole text at the proofreading stage.

A short not-exahustive list of issues is:

-  at page 3 the authors employ the term "lithization" (which is not written correctly in the text, by the way), whereas at page 4 the term "lithiation" appears, which I had never seen before.

- Page 16. "Interesting to note, that an example of the effective use of Raman spectroscopy is the detection of carbon dust in the NSTX tokamak with plasma materials made of graphite and tiles from a carbon fiber composite.". The subject "It is" is missing at the beginning of the sentence. 

- page 30. Beginning of section 12. "It is shown that the technique for studying CDx hydrocarbon films developed in the recent one-two decades and consisting of a dozen experimental techniques that have shown that the properties of the films are, in fact, a "passport" of the processes of their formation under the action of the plasma conditions of the T-10 tokamak".

The sentence starts with a subject and ends with another.
